# The tricellular vertex-specific adhesion molecule Sidekick facilitates polarised cell intercalation during *Drosophila* axis extension

**Tara M. Finegan**[1‡], **Nathan Hervieux**[1‡], **Alexander Nestor-Bergmann**[1], **Alexander G. Fletcher**[2,3], **Guy B. Blanchard**[1], **Bénédicte Sanson**[1]*

**1** Department of Physiology, Development and Neuroscience, University of Cambridge, Cambridge, United Kingdom, **2** School of Mathematics and Statistics, University of Sheffield, Sheffield, United Kingdom, **3** Bateson Centre, University of Sheffield, Sheffield, United Kingdom

‡ These authors share first authorship on this work.
* bs251@cam.ac.uk

## Abstract

In epithelia, tricellular vertices are emerging as important sites for the regulation of epithelial integrity and function. Compared to bicellular contacts, however, much less is known. In particular, resident proteins at tricellular vertices were identified only at occluding junctions, with none known at adherens junctions (AJs). In a previous study, we discovered that in *Drosophila* embryos, the adhesion molecule Sidekick (Sdk), well-known in invertebrates and vertebrates for its role in the visual system, localises at tricellular vertices at the level of AJs. Here, we survey a wide range of *Drosophila* epithelia and establish that Sdk is a resident protein at tricellular AJs (tAJs), the first of its kind. Clonal analysis showed that two cells, rather than three cells, contributing Sdk are sufficient for tAJ localisation. Super-resolution imaging using structured illumination reveals that Sdk proteins form string-like structures at vertices. Postulating that Sdk may have a role in epithelia where AJs are actively remodelled, we analysed the phenotype of *sdk* null mutant embryos during *Drosophila* axis extension using quantitative methods. We find that apical cell shapes are abnormal in *sdk* mutants, suggesting a defect in tissue remodelling during convergence and extension. Moreover, adhesion at apical vertices is compromised in rearranging cells, with apical tears in the cortex forming and persisting throughout axis extension, especially at the centres of rosettes. Finally, we show that polarised cell intercalation is decreased in *sdk* mutants. Mathematical modelling of the cell behaviours supports the notion that the T1 transitions of polarised cell intercalation are delayed in *sdk* mutants, in particular in rosettes. We propose that this delay, in combination with a change in the mechanical properties of the converging and extending tissue, causes the abnormal apical cell shapes in *sdk* mutant embryos.

## Introduction

Vertices are the points of contact between three or more cells in an epithelium. Epithelial vertices have known specialised junctions at the level of occluding junctions [1] (see Fig 1A). In

**Data Availability Statement:** Data for all graphs in the paper are stored in the University of Cambridge Apollo repository with the permanent DOI: https://

doi.org/10.17863/CAM.44798 Simulations code
and source data are stored in the GitHub repository
at https://github.com/Alexander-Nestor-Bergmann/
GBE_vertex_model_sdk

**Funding:** This work was supported by two
successive Wellcome Trust Investigator Awards
(099234/Z/12/Z and 207553/Z/17/Z) and a BBSRC
standard grant (BB/J010278/1) to BS; a University
of Cambridge ISSF Junior Interdisciplinary
Fellowship and then a Herchel Smith Postdoctoral
Fellowship to ANB; a University of Sheffield Vice-
Chancellor's Fellowship to AGF; and a University of
Cambridge Studentship from the Wellcome Trust
Developmental Biology PhD program to TMF. The
funders had no role in study design, data collection
and analysis, decision to publish, or preparation of
the manuscript.

**Competing interests:** The authors have declared
that no competing interests exist.

**Abbreviations:** AJ, adherens junction; AP,
anteroposterior; aPKC, Atypical protein kinase C;
CPTI, Cambridge Protein Trap Insertion; Dlg, Discs
large; DV, dorsoventral; Fn, fibronectin; FRT/FLP,
yeast site-specific recombination system; Gap43,
growth-associated protein 43; GBE, germband
extension; GFP, green fluorescent protein; hs, Heat
shock; Ig, immunoglobulin; ILDR,
Immunoglubulin-like domain-containing receptor;
LSR, Lipolysis-stimulated lipoprotein receptor;
MAGI, Membrane-associated guanylate kinase
inverted; nls, nuclear localisation signal; PDZ,
postsynaptic density protein 95/Dlg protein/Zonula
occludens 1 domain; RFP, red fluorescent protein;
Sdk, Sidekick; SIM, Structured Illumination
Microscopy; Sqh, spaghetti-squash; tAJ, tricellular
adherens junction; tSJ, tricellular septate junction;
tTJ, tricellular tight junction; YFP, yellow
fluorescent protein.

vertebrates, a protein complex containing tricellulin and the angulins Immunoglubulin-like domain-containing receptor (ILDR)1, ILDR2, and LSR Lipolysis-stimulated lipoprotein receptor (LSR) forms tricellular tight junctions (tTJs) [1–3]. Absence of tTJs causes loss of the epithelial barrier function and is associated with health defects such as familial deafness [4, 5]. In invertebrates, septate junctions (the functional homologue of vertebrate tight junctions) also harbour specialised proteins at tricellular junctions, namely the transmembrane proteins Gliotactin and Anakonda, both required for epithelial barrier function, and the more recently identified protein M6 [1, 6–9].

In addition to their role in epithelial barrier function, tricellular junctions might be important for the mechanical integrity of epithelia [10]. During animal development, dynamic cell behaviours drive the morphogenetic remodelling of epithelial tissues. For example, polarised cell intercalation, cell division, and cell extrusion require the remodelling of cell–cell contacts, posing a conformational problem at tricellular junctions that has started to be addressed [10–12]. Tricellular vertices might also be sites for tension sensing and transmission, which could be important for epithelial remodelling [1, 13–17]. Force sensing and transmission have been more often associated with adherens junctions (AJs) rather than occluding junctions [18]. While proteins are known to localise specifically at tricellular occluding junctions, it remains to be determined if a similarly specialised structure is present at the level of AJs at tricellular vertices. In *Drosophila*, many proteins are known to enrich at tricellular vertices at the level of AJs [19–22]. However, only one so far has been found to be specifically localised there, the *Drosophila* immunoglobulin (Ig) superfamily adhesion protein Sidekick (Sdk) [19]. To our knowledge, there are no other proteins known in invertebrate or vertebrate animals yet with such specific tricellular AJ (tAJ) localisation (see Fig 1A).

The *sdk* gene was first identified in *Drosophila* as required for normal ommatidial differentiation [23]. The vertebrate homologues *Sidekick-1* (*Sdk-1*) and *Sidekick-2* (*Sdk-2*) regulate lamina-specific connectivity during retinal development, with Sdk-2 being important for the circuitry detecting differential motion [24, 25]. In *Drosophila*, Sdk has also been demonstrated to establish the visual motion detection circuit [26]. Besides their neuronal functions, Sdk proteins have been implicated in podocyte function in the kidney [27], but to our knowledge, a localisation at tricellular vertices has not been reported. All Sdk proteins share large extracellular domains composed of Ig and fibronectin (FN) domains and a short cytoplasmic domain with a conserved C-terminal hexapeptide sequence predicted to bind postsynaptic density protein 95/Discs large (Dlg) protein/Zonula occludens 1 (PDZ) domains (see S1A and S1B Fig) [28]. Vertebrate Sdk molecules bind the Membrane-associated guanylate kinase inverted (MAGI) protein via this conserved C-terminal motif, which is important for both renal and neuronal functions [27, 28]. Structure studies of Sdk-1 and Sdk-2 ectodomains show that they form homophilic dimers in *cis-* and in *trans-* [29, 30].

In this paper, we address a possible role of Sdk in epithelial morphogenesis and investigate its function in an epithelium known to undergo active remodelling, the *Drosophila* germband. *Drosophila* axis extension (called germband extension, GBE) is a paradigm for convergence and extension movements in epithelia [31]. Cells undergo polarised cell intercalation under the control of the anteroposterior (AP) patterning genes [32]. Planar polarisation of the actomyosin cytoskeleton and AJs drives the rearrangement of groups of four cells (called T1 transitions) or more (rosettes), leading to tissue extension [33–36]. The cell biology of junctional remodelling has been extensively studied for bicellular contacts [20, 31, 37–39]. The role of vertices, however, has been the focus on only one study so far [40]. In parallel to the intrinsic forces of polarised cell intercalation, extrinsic forces also contribute to tissue extension [41]. These are caused by the invagination of the endoderm at the posterior of the germband, pulling on the extending tissue [42, 43]. The extensive knowledge available about germband

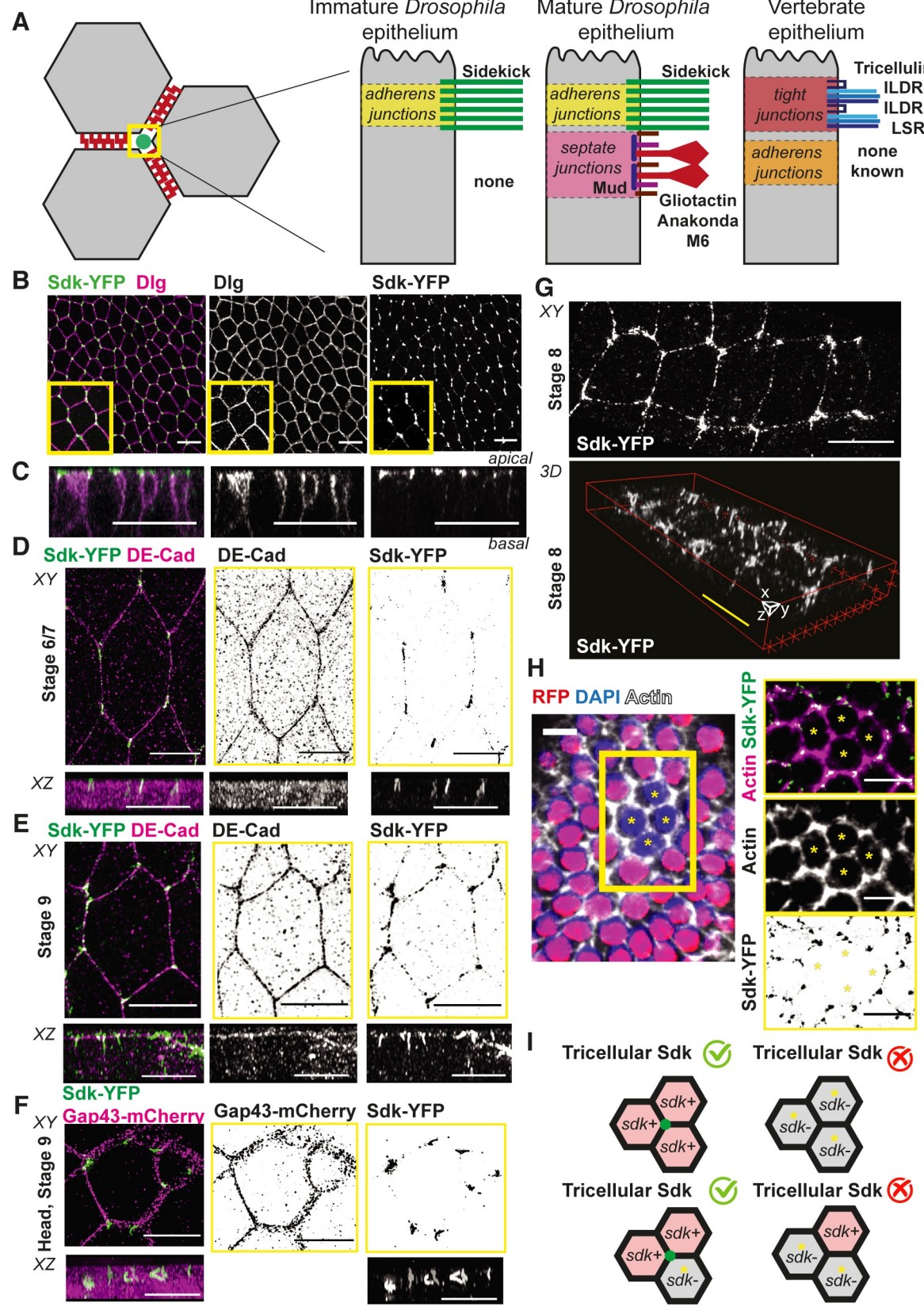

**Fig 1. Sdk is a novel component of apical vertices in epithelia.** (A) Cartoon summarising the geometry of tricellular vertices and the proteins known to localise specifically there in epithelia of different types. (B,C) Immunostaining of fixed ventral ectoderm of stage 7 *Drosophila* embryos for Sdk-YFP and the lateral marker Dlg. (B) Maximum projection. Scale bar = 5 μm. (C) Z-reconstruction based on confocal slices taken from confocal stack in B. Scale bar = 20 μm. (D–G) Super-resolution SIM imaging of fixed embryos immunostained with Sdk-YFP and either DE-Cad to show the position of AJs (D,E) or Gap43-mCherry (F) to label the whole membrane. This reveals the localisation of Sdk in 'strings' at apical vertices in embryonic epithelia. Images are maximum projection (labelled XY) and z-reconstruction (labelled XZ) from z-slices of the apical side of the cells. Scale bars, apical views (top panels) = 5 μm; lateral views (bottom panels) = 2 μm. (G) 3D reconstruction to show the z-component of the strings. Scale bars = 5 μm. (H) Example of an sdk[Δ15] clone (absence of nls-RFP signal) in the follicular epithelium (stage 6 egg chamber) stained for Sdk-YFP and actin phalloidin. Right panels show a close-up of the four-cell clone (asterisks) with individual channels on the left to show Sdk-YFP and actin, respectively, and the merge. Scale bars = 20 μm. (I) Cartoon summarising the results of the clone experiment shown in H. Vertex localisation of Sdk requires Sdk proteins in at least two of the three cells forming a vertex. AJ, adherens junction; DE-Cad, DE-cadherin; Dlg, Discs large; Gap43, growth-associated protein 43; ILDR, Immunoglublulin-like domain-containing receptor; LSR, Lipolysis-stimulated lipoprotein receptor; nls, nuclear localisation signal; RFP, red fluorescent protein; Sdk, Sidekick; SIM, Structured Illumination Microscopy; YFP, yellow fluorescent protein.

extension makes it an excellent system to search for a role of the tricellular vertex protein Sidekick. Moreover, the *Drosophila* embryo at gastrulation does not have septate junctions yet, so potential redundant mechanical roles between those and AJs will be absent.

## Results

### The Ig superfamily adhesion molecule Sdk localises specifically at cell vertices at the level of AJs in *Drosophila* epithelia

We identified Sdk as a marker of tAJs in early embryos in a screen of the Cambridge Protein Trap Insertion (CPTI) collection of yellow fluorescent protein (YFP) traps [19]. The three independently generated CPTI protein traps in *sdk*, all located at the N-terminus of the protein (S1A and S1B Fig) [44], showed the same vertex-specific localisation (S1C Fig), so we used one of them, *sdk-YFP*[CPTI000337], for further characterisation (shortened as Sdk-YFP below). We surveyed other epithelia to ask whether Sdk also localised at apical vertices there. In the large majority of epithelia, Sdk-YFP localises at vertices at the level of AJs as in the early embryo [19] (Fig 1B and 1C and S1 Table and S2 Fig). For example, Sdk is at tAJs in the amnioserosa, a squamous epithelium in embryos; in the larval wing disc, a pseudostratified epithelium; and in the early adult follicular cells, a cuboidal epithelium. Vertex localisation of Sdk is found both in immature epithelia without septate junctions (early embryo, amnioserosa, and follicular layer) and in mature epithelia with septate junctions (S1 Table). This suggests that the localisation of Sdk at tAJs is independent of the presence of tricellular septate junctions (tSJs). There are a few notable exceptions to the tricellular localisation of Sdk in epithelia: in third instar salivary glands and in the follicular epithelium after stage 7, Sdk is all around the membrane (S1 Table and S2 Fig), whereas during GBE, Sdk-YFP appears planar-polarised at bicellular contacts in addition to tricellular localisation (see below and Fig 1B and S1F Fig). Also, in the adult midgut, Sdk is not detected, consistent with the midgut's atypical apicobasal polarity [45] (S2F Fig and S1 Table). From this survey of many epithelia, we conclude that Sdk is a resident protein of tAJs in *Drosophila* epithelia, the first of its kind.

Next, we used Structured Illumination Microscopy (SIM) to examine Sdk localisation in fixed early embryos at a resolution higher than conventional confocal microscopy (see Materials and Methods). With this higher resolution (about 100 nm in XY and 125 nm in Z), the Sdk-YFP signal resolves as a string-like object at vertices (Fig 1D–1G). The Sdk-YFP strings are seen at all stages and in all regions of the early embryo, in epithelia that are remodelling such as the ventrolateral ectoderm (Fig 1D, 1E and 1G), and in more inactive epithelia such as the head ectoderm (Fig 1F). This suggests that the localisation of Sdk-YFP in strings or plaques is a general feature of the epithelium. The strings are continuous at vertices, and co-staining

with E-Cadherin (Fig 1D and 1E) or Atypical protein kinase C (aPKC), a marker of the apical domain (S1D Fig), shows that Sdk strings extend a little beyond the E-Cadherin belt both apically and basally (S1E Fig). Sdk has a very long extracellular domain (>2,000 aa) and is tagged with YFP at the end of this domain (S1B Fig). The length of the strings, around 2 μm, suggests the formation of an assembly of Sdk proteins containing YFP in the extracellular space at apical vertices. Because the YFP tag is at the end of the extracellular domain of Sdk, a terminal fragment could be forming the strings alone. To rule this out, we co-stained Sdk-YFP–expressing embryos with an antibody raised against the intracellular domain of Sdk (S1B Fig)[26]. The same string-like localisation was observed, showing that the strings are likely made of the whole of the Sdk protein (S1F and S1G Fig).

The presence of string-like structures suggests that Sdk proteins form specialised assemblies at vertices. Because vertebrate Sdk proteins are known to bind homophilically, we asked how many cells expressing Sdk are required for Sdk vertex localisation. Because of the lack of cell divisions in the ectoderm at gastrulation, mosaic analysis cannot be performed using the yeast site-specific recombination system FRT/FLP [46], so we generated mosaics in the follicular cell layer of female ovaries. X chromosomes bearing either FRT (recombinase recognition site), nuclear localisation signal-red fluorescent protein (nls-RFP), and sdk-YFP or FRT and the mutation $sdk^{A15}$ [26] were constructed and mosaics produced using heat shock (hs)-Flp (recombinase under hs control) (see Materials and Methods). Heat shocks were timed to generate clones in the follicular epithelium, and tricellular vertices with one, two, or three mutant cells were examined for Sdk-YFP fluorescence (Fig 1H and 1I). We find that tricellular vertices with three mutant *sdk* cells do not have Sdk-YFP signal at vertices, showing that there is no perdurance of the protein in the mutant clones. Whereas vertices contributed by one mutant and two wild-type cells are positive for Sdk-YFP, we find that vertices contributed by two mutant cells and one wild-type are not (Fig 1H and 1I). This indicates that two cells contributing Sdk are sufficient for localising Sdk at tricellular vertices.

## Sdk localisation changes when junctions are remodelled during axis extension

As mentioned above, Sdk-YFP appears planar-polarised during convergence and extension of the *Drosophila* germband at gastrulation (Fig 1B and S1F Fig). During this morphogenetic movement, vertices are remodelled during polarised cell intercalation, opening the possibility that the planar polarisation of Sdk was linked to this remodelling. To test this, we made movies of embryos labelled with Sdk-YFP and E-Cadherin-mCherry (to label the AJs). We consistently observed a different behaviour of Sdk at shortening versus elongating junctions during T1 transitions (Fig 2A and 2B). At some point during junction shortening, Sdk-YFP loses its sharp punctate localisation at vertices and apparently becomes distributed all along the shortening junction (time points 40 to 120 seconds in example shown in Fig 2A) until it sharpens again into a single punctum at the four-cell intermediate (time points 140 to 160 seconds, Fig 2A). In contrast, when the new junction begins to grow, the single Sdk-YFP punctum appears to immediately split into two sharp puncta flanking the elongating junction (time points 160 to 180 seconds, Fig 2A).

We examined further this differential localisation of Sdk in our super-resolution data. In non-intercalating cells, Sdk-YFP strings tend to be aligned parallel to the apicobasal axis, adopting a 'vertical' configuration (see, for example, in stage 9 embryos, Fig 1E and 1F). In contrast, in intercalating cells, Sdk-YFP strings tend to be more planar: some strings are completely planar ('planar' configuration), whereas others are vertical, then planar ('step' configuration) (see examples in Fig 2C–2E). Using 3D reconstruction of the super-resolution data

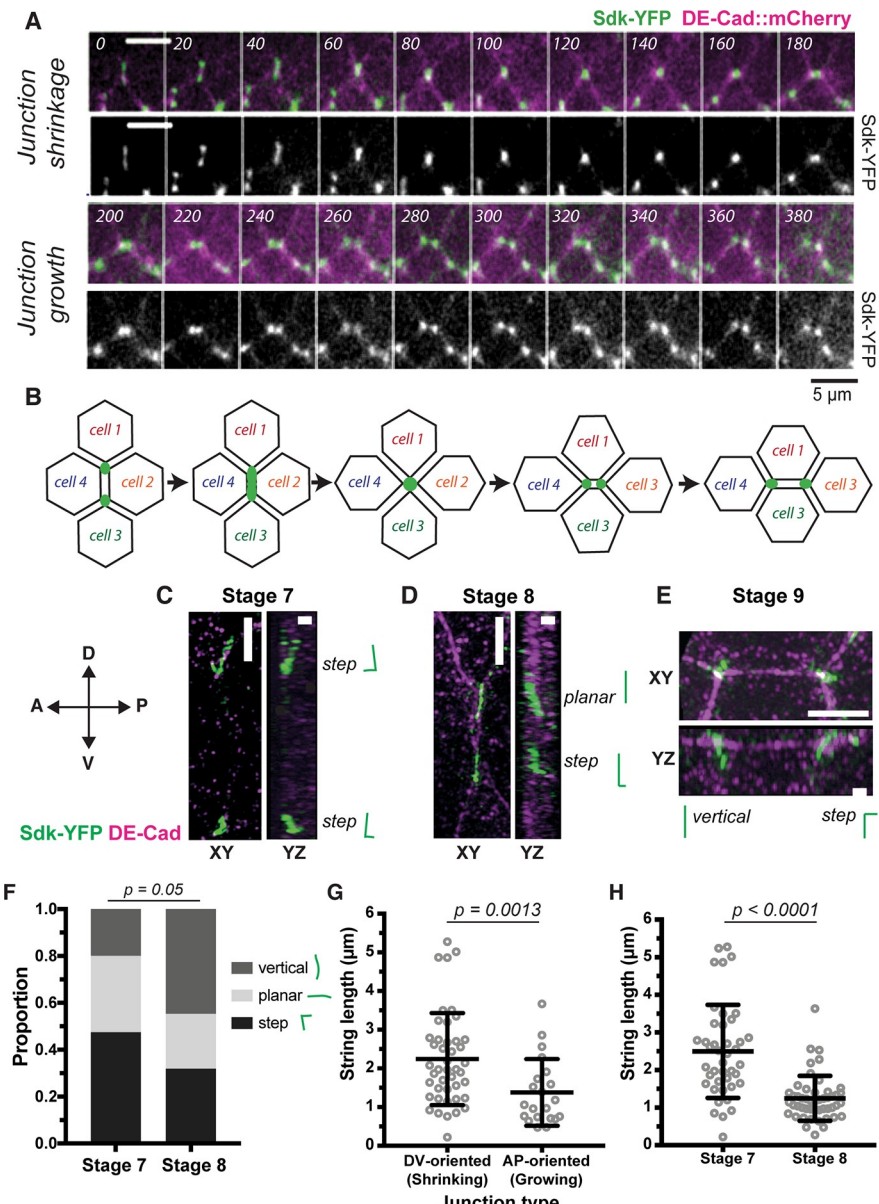

**Fig 2. Sdk localises differently at shortening versus elongating junctions during polarised cell intercalation.** (A,B) Sdk localisation during a T1 transition imaged over 20 minutes in live embryos labelled with Sdk-YFP and DE-Cad-mCherry$^{KI}$. Time is indicated in seconds since the start of the intercalation event. Each image is a maximum intensity projection over 3 z-slices spanning 1.5 μm. This movie is representative of behaviour found in all of $n = 9$ complete intercalation events, $n = 8$ junction shrinkages, and $n = 6$ junction growths. (B) Cartoon illustrating the behaviour of Sdk-YFP shown in A. (C–H) Analysis of Sdk-YFP string localisation at shortening and elongating junctions by super-resolution SIM. Embryos are fixed and stained for GFP and E-Cad. Scale bars = 1 μm. (C–E) Representative SIM super-resolution images of DV-oriented junctions at late stage 6 (C) and at stage 7 (D) and of an AP-oriented junction at stage 8 (E). Orientation is within 20° of AP or DV axis. For each example, the string classification used in F is shown. (F) Quantification of string morphologies based on 3D reconstructions in stage 7 and stage 8 embryos. Morphologies where divided into three classes: vertical, planar, and step-like (stage 7: total $n = 40$, step = 19, planar = 13, vertical = 8; stage 8: total $n = 47$, step = 15, planar = 11, vertical = 21). Statistical significance calculated by chi-squared test. (G) Quantification of string lengths at shrinking versus growing junctions (defined by their orientation within 20° of AP or DV embryonic axis, respectively; shrinking: $n = 44$, growing: $n = 21$). Statistical significance calculated by Mann–Whitney test. (H) Quantification of string lengths at stage 7 versus stage 8 (stage 7: $n = 42$, stage 8: $n = 49$). Statistical significance calculated by Mann–Whitney test. Data for graphs F–H can be found at https://doi.org/10.17863/CAM.44798. AP, anteroposterior; DE-Cad, DE-Cadherin; DV, dorsoventral; GFP, green fluorescent protein; Sdk, Sidekick; SIM, Structured Illumination Microscopy; YFP, yellow fluorescent protein.

(see Materials and Methods), we systematically classified the configurations of strings in stage 7 and stage 8 embryos. At stage 7, when cells are intercalating actively, 80% of the strings have a 'planar or 'step' configuration, this proportion decreasing to about 50% at stage 8, when cell intercalation starts to decrease (Fig 2F). 3D reconstruction allows one to measure the length of Sdk-YFP strings accurately, and we find that strings are longer at dorsoventral (DV)-oriented junctions compared to AP-oriented junctions (Fig 2G). Strings are also longer at stage 7 compared to stage 8 (Fig 2H). Together, these quantifications support the notion that Sdk-YFP strings become longer and more planar during shrinkage of DV-oriented junctions, whereas the strings are shorter and more vertical when AP-oriented junctions are growing. We also infer that the planar string patterns of DV-oriented junctions in our super-resolution data are likely to correspond to the continuous distribution of Sdk-YFP observed at shortening junctions in the live data at lower resolution (Fig 2A and 2B).

Based on the above live and fixed data, we propose that Sdk localisation during a T1 transition follows the sequence illustrated in Fig 2C–2E. Two possible explanations are possible for this change in Sdk localisation: Sdk either moves to bicellular contacts at shortening junctions or, alternatively, remains at tricellular contacts, but cells form protrusions extending towards the shortening junctions (S1H and S1I Fig, and see Discussion). Because the increase in resolution with SIM is moderate, we were unable to distinguish between these two possibilities. We conclude that the localisation of Sdk is different between shortening and growing junctions, suggesting that Sdk may play a role in polarised cell intercalation.

## In intercalating cells, rosette centres contain separable tricellular vertices marked by Sdk

Next, we examined the localisation of Sdk during rosette formation (Fig 3). Rosettes are observed in the germband when several contiguous DV-oriented junctions shorten together, merging into an apparently single vertex [35]. It is not known whether each rosette centre really represents a single junctional vertex structure or not. To address this, we made movies of Sdk-YFP embryos also labelled with growth-associated protein 43 (Gap43)-Cherry to label all cell membranes. Live imaging suggests that rosette centres are in fact made of several puncta of Sdk-YFP, which move relative to each other in a dynamic fashion during rosette cell rearrangements (Fig 3A). The sequence of shortening and elongation of very short junctions between Sdk-YFP puncta suggested that tricellular vertices marked by Sdk might remain separated in rosette centres. This has implications for how we understand polarised cell intercalation because this suggests that rosettes might resolve through successive T1 transitions.

To test this, we examined rosette centres in our super-resolution data (Fig 3B). Maximal projections (for example, see the XY projection in Fig 3B) are unable to reveal whether Sdk-YFP strings are continuous or separate. Thus, as above, we used 3D reconstruction to follow the path of the strings at rosette centres (Materials and Methods). This analysis revealed that several strings are always observed in the middle of rosettes and they are not in contact with each other (see reconstruction in Fig 3B). This indicates that rosette centres are composed of separable tricellular apical vertices marked by Sdk. We also examined the configuration of rosette centres below the AJs, marking the whole membrane (for example, using concavalin A, S3 Fig). As observed by others [47], we find that the cell connectivity can change significantly within the apical-most 3 μm. In the case shown in S3 Fig, whereas three distinct Sdk strings are present in the apical-most portion of the rosette centre, a single punctum of Sdk is found 2 μm below, where the connectivity of the cells differs. We conclude that Sdk strings corresponds to the apical-most junctional conformation and that during junctional exchange at the level of AJs, single junctional vertex intermediates are not usually formed between more than four

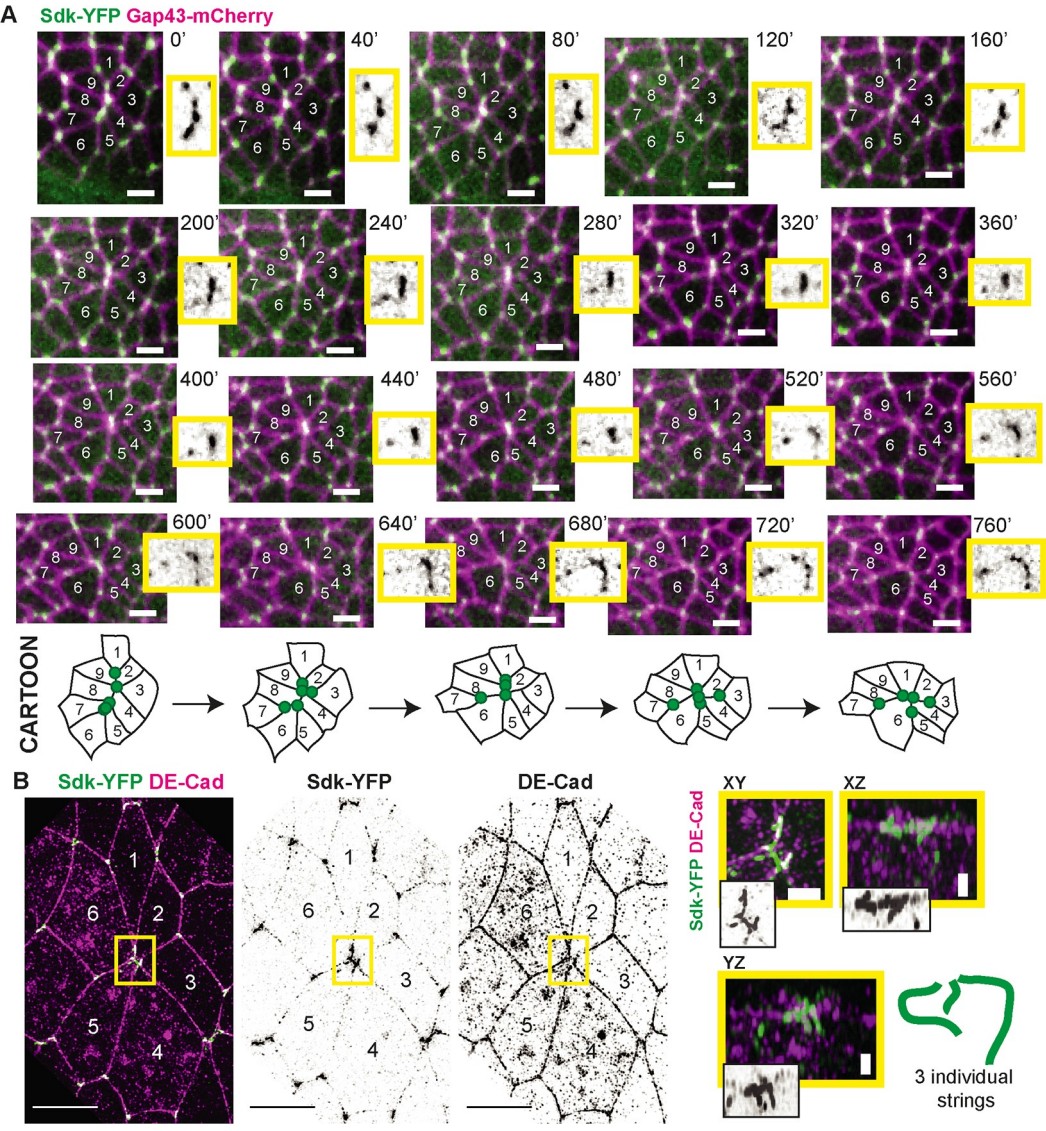

**Fig 3. tAJs marked by Sdk-YFP are separate at rosette centres.** (A) Sdk localisation during rosette formation imaged over 15 minutes in live embryos labelled with Sdk-YFP and Gap43-mCherry. Time is indicated in seconds since start of the intercalation rosette. Each image is a maximum intensity projection over 3 z-slices spanning 1.5 μm. Movie is representative of behaviour found in all of $n$ = 6 full rosette-like intercalation events. Scale bars = 5 μm. Close-up images of the rosette centre are shown in yellow boxes for the Sdk-YFP channel. Cartoon below illustrates the dynamics of the Sdk-YFP puncta seen in the movie. (B) Sdk-YFP string localisation at a rosette centre involving six cells, imaged by super-resolution SIM. The image is from a stage 8 embryo fixed and stained for GFP and DE-Cad. Maximum projection over 12 slices = 1.5 μm. Close-ups of the rosette centre with different projections are shown in yellow boxes to demonstrate that three distinct strings can be resolved. Cartoon shown to interpret images. Scale bars for the main SIM panels are 5 μm and for the close-ups 1 μm. DE-Cad, DE-Cadherin; Gap43, growth-associated protein 43; GFP, green fluorescent protein; Sdk, Sidekick; SIM, Structured Illumination Microscopy; tAJ, tricellular adherens junction; YFP, yellow fluorescent protein.

cells. This suggests that intercalation events forming rosettes occur through separable T1-like events.

## Loss of sdk causes abnormal cell shapes during GBE

To investigate a possible role of Sdk in GBE, we made movies of embryos homozygous for the $sdk^{MB5054}$ null mutant [26] and carrying E-cadherin-GFP [48] to label apical cell contours.

Because *sdk* loss-of-function mutations are viable [23], these embryos are devoid of both maternal and zygotic contributions for Sdk. Consistent with a role in GBE, we observe an abnormal cell shape phenotype in *sdk* null mutants during GBE, with distinct differences in the geometry and topology of the apical planar cellular network compared to the wild type; in particular, many cells have a less regular and more elongated polygonal shape (Fig 4A and 4B).

To describe these phenotypes quantitatively, we acquired five wild-type and five *sdk* movies of the ventral side of embryos over the course of GBE. We then segmented the cell contours, tracked cell trajectories through time, and synchronised movies within and between each genotype group, as previously [41, 43, 49] (Materials and Methods). To allow comparisons between wild-type and *sdk* embryos, we defined the beginning of GBE (time 0) using a given threshold in the rate of tissue extension (see Materials and Methods, S4A and S4B Fig and S1 and S2 Movies). The total number of ventral ectoderm cells in view and analysed increased from start to end of GBE, from about 500 cells to above 2,000 cells for both wild-type and *sdk* embryos (S4C and S4D Fig).

We first analysed the anisotropy in cell shapes and their orientation in the course of GBE (Fig 4C). The eccentricity of ellipses fitted to the apical cell shapes is used as a measure of cell shape anisotropy (see Materials and Methods). The orientation of the ellipse's major axis relative to the embryonic axes gives the cell orientation. At the beginning of GBE, ectodermal cells are elongated in DV because the tissue is being pulled ventrally by mesoderm invagination [41, 43] (Fig 4D and S5A–S5C Fig). In wild-type embryos, the cells then become progressively isotropic as the ectoderm extends, as we showed previously [41]. In contrast, in *sdk* embryos, the cell shapes become briefly isotropic and then become anisotropic again, this time along the AP axis (Fig 4D and S5B and S5C Fig). This anisotropy in the AP direction could be due to cells being longer in AP, thinner in DV, or both. To distinguish between these possibilities, we measured the cell lengths along AP or DV (Fig 4E and 4F and S5D and S5E Fig). We found that both cell lengths are significantly different in *sdk* mutants compared to the wild type, with cells being shorter in DV and also, but more moderately, longer in AP.

Because the AP and DV cell lengths described above are a projection of ellipses fitted to the cell shapes, we also looked directly at the length of the cell–cell interfaces in the course of GBE (Fig 4G and 4H and S5F–S5I Fig). We classified cell interfaces as being AP- or DV-oriented based on their angles with the embryonic axes (see Materials and Methods). Mirroring the cell length results, we find AP-oriented interfaces get a little longer and the DV-oriented interfaces shorter in *sdk* compared to wild-type embryos in the course of GBE.

Together, our cell shape quantifications demonstrate that overall, *sdk* cells become shorter in DV and, to a lesser extent, longer in AP in the course of GBE, consistent with our initial qualitative observation of many more elongated cells in *sdk* mutants. We hypothesised that this cell shape phenotype could be a consequence of a defect in polarised cell intercalation, which would, in turn, modify cell topologies.

## Tears in the apical cortex persist in *sdk* mutants during polarised cell intercalation

A possibility is that Sdk is required for normal polarised cell intercalation through mediating homophilic adhesion or anchoring the actomyosin cytoskeleton when cells rearrange. Supporting this notion, we noticed apical discontinuities in the converging and extending epithelium in *sdk* mutants labelled with E-Cadherin-GFP[KI] and Myosin II-Cherry (Fig 5 and S6 Fig). These apical tears or gaps are lined by the actomyosin cortex and usually associated with a depletion in E-Cadherin (Fig 5A and 5C). In an example in which E-Cadherin was still present around a small circular gap, following the signal more basally showed that the gap had closed

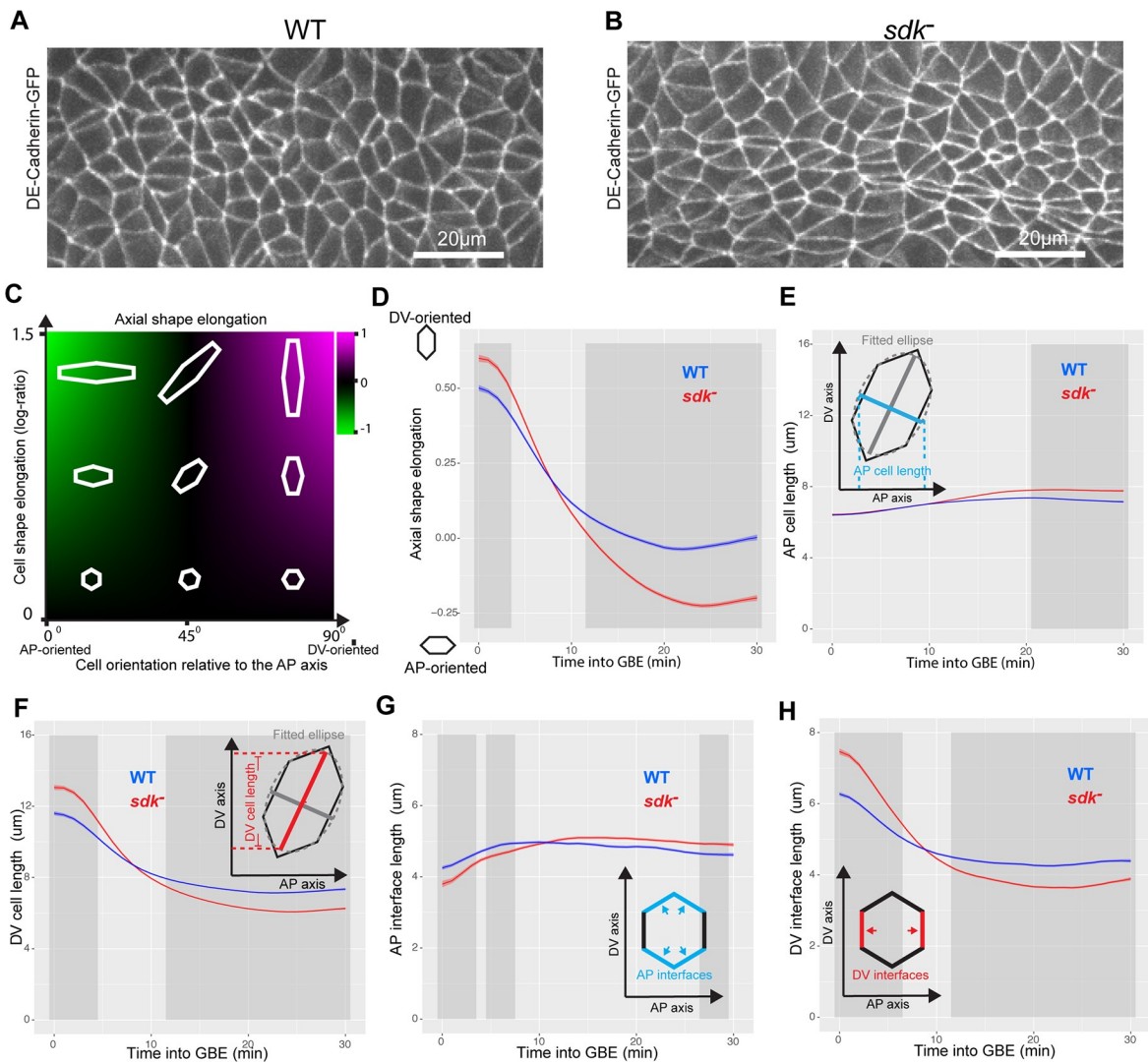

**Fig 4. Cell shapes are more anisotropic in *sdk* mutants versus the WT during GBE.** (A,B) Movie frame of ventral ectoderm at 30 mins into GBE from representative WT (A) and *sdk* (B) movies, labelled with E-cadherin-GFP. (C,D) Measurement of cell shape anisotropy and orientation (see also S5A–S5C Fig). Cell shape anisotropy is calculated as the log ratio of the principal axes of best-fit ellipses to tracked cell contours. An isotropic cell shape (a circle) will have a log-ratio value of 0 and a very elongated cell a value of over 1. Cell orientation is given by the cosine of the angular difference between the ellipse's major axis and the DV embryonic axis. Negative values indicate cells that are elongated in the AP axis, positive values in the DV axis. Cell shape anisotropy and orientation measures are then multiplied together to give a composite measure (termed 'axial shape elongation') of how elongated cells are in the orientation of the embryonic axes (Materials and Methods). (D) Axial shape elongation measure (y-axis) for the first 30 mins of GBE (x-axis) for WT and *sdk* embryos. In this graph and hereafter, the ribbon's width indicates the within-embryo confidence interval, and the dark grey shading indicates a difference ($p < 0.05$) (Materials and Methods). (E–F) Measures of AP and DV cell lengths in WT and *sdk* embryos (see also S5D and S5E Fig). Cell shape ellipses are projected onto AP and DV axes to derive a measure of cell length in each axis. (G,H) Evolution of AP-oriented and DV-oriented cell–cell interface lengths (y-axis) as a function of time in GBE (x-axis) (see also S5F and S5I Fig). Tracked cell–cell interfaces are classified as AP- or DV-oriented according to their orientation relative to the embryo axes. Data for graphs D–H can be found at https://doi.org/10.17863/CAM.44798. AP, anteroposterior; DV, dorsoventral; GBE, germband extension; GFP, green fluorescent protein; Sdk, Sidekick; WT, wild type.

already 1 μm below the AJs (Fig 5B). These apical tears seemed particularly prevalent and larger at the centres of rosettes, forming oval structures bordered by Myosin II, as shown in super-resolution images in Fig 5C.

Next, we systematically looked for these cortical discontinuities, comparing movies of *sdk* and wild-type embryos labelled with E-Cadherin-GFP[KI] and Myosin II-Cherry. Unexpectedly,

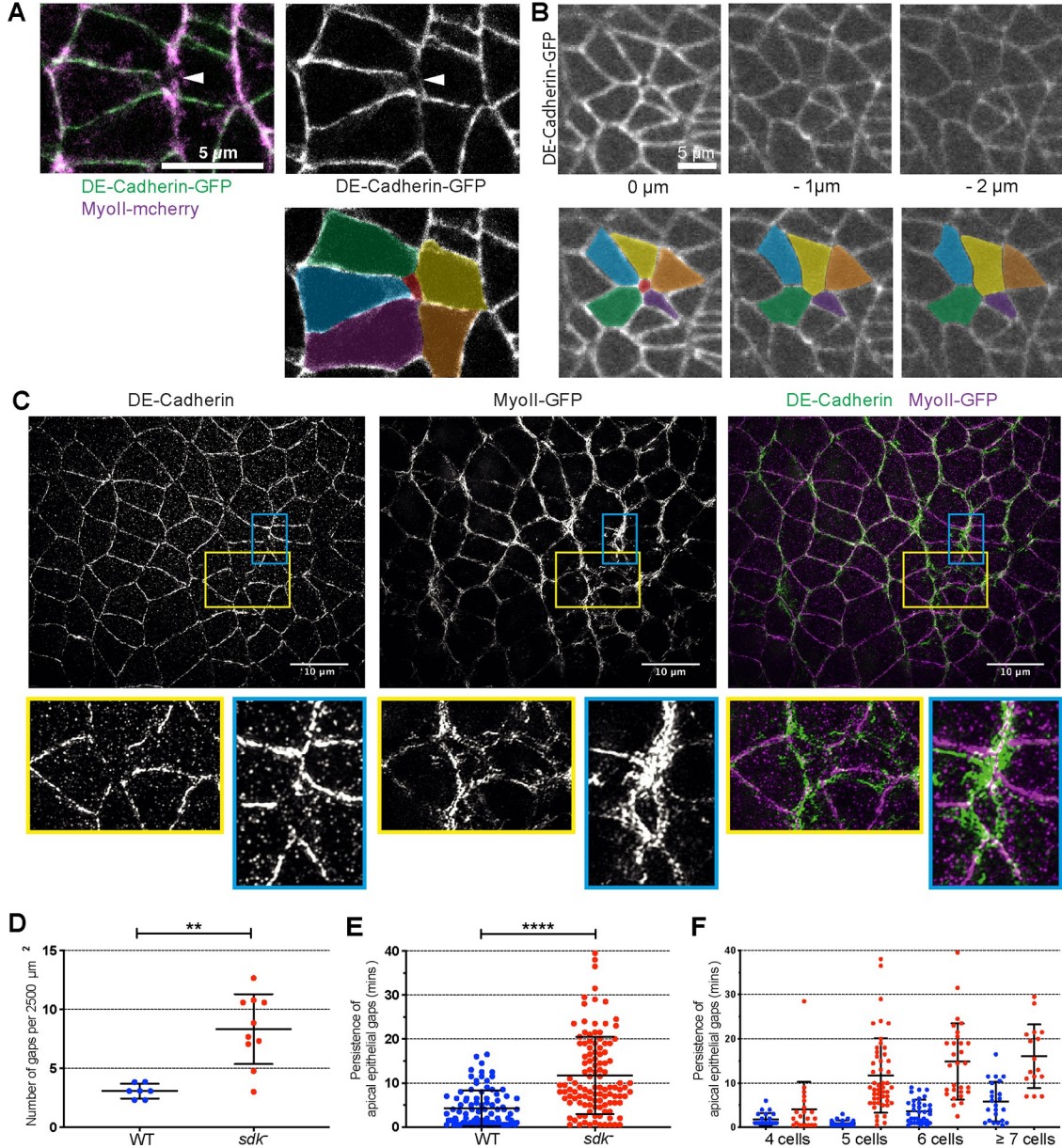

**Fig 5. Apical adhesion is disrupted during polarised cell intercalation in s*dk* mutants.** (A) A single z-frame at the level of AJs showing a gap or tear in the cortex at a presumed rosette centre in an *sdk* mutant embryo. Left panel shows merge between DE-Cad-GFP and Sqh-mCherry (shortened as MyoII-mCherry) signals, right panel, DE-Cad-GFP channel only. In the bottom panel, the different cells have been coloured to highlight the apical gap in the middle. (B) Single z-frames at different positions along the apicobasal axis of an apical gap in an *sdk* mutant embryo (from movie shown in S6A Fig) at the level of AJs (0 μm) and 1 and 2 μm below. The gap present at the level of AJs is closed in the planes basal to the AJs. Bottom panels show colourised cells, highlighting the apical gap in the apical-most z-slice. (C) Fixed and stained *sdk* mutant embryo against DE-Cad and GFP (to reveal MyoII-GFP), imaged by SIM super-resolution microscopy at stage 7. Each image is a maximum intensity projection over 3 μm at the level of AJs in the ventral ectoderm. Regions bounded by yellow and blue lines show discontinuities in E-Cad signal, indicating holes in apical adhesion, and are shown below as close-ups. (D–F) Apical gaps quantifications in WT and *sdk* mutant movies as shown in A, B. (D) Quantification of the number of gaps found at the level of AJs, normalised to a given area (2,500 μm²) of the ectoderm. One to two regions (embryo sides) were quantified per movie: WT, $n = 7$, from seven embryos; *sdk* mutant $n = 10$, from eight embryos; Mann–Whitney, *p*-value = 0.0018. (E) Quantification of how long apical gaps persist in the tissue. (F) Quantification of how long apical gaps persist as a function of the number of cells present at the gap's border. We detected gaps where four to seven cells and more meet. For both E and F, the number of gaps quantified was $n = 92$ for WT and $n = 115$ for *sdk* mutant. In D, Mann–Whitney, *p*-value < 0.0001. Data for graphs D–F can be found at https://doi.org/10.17863/CAM.44798. AJ, adherens junction; DE-Cad, DE-Cadherin; GFP, green fluorescent protein; MyoII, Myosin II; Sdk, Sidekick; SIM, Structured Illumination Microscopy; Sqh, spaghetti-squash; WT, wild type.

we also found apical gaps in wild-type embryos in the course of GBE, which to our knowledge has never been reported. However, compared to the wild type, apical gaps are more numerous in *sdk* mutants and also persist in the epithelium for much longer (Fig 5D and 5E and S6A Fig). In both the wild type and *sdk* mutants, the apical gaps appear associated with groups of cells undergoing polarised rearrangements. Gaps forming where four cells meet are likely to represent single T1 transitions and are the most transient (Fig 5F). Apical holes forming where five cells or more meet are likely to correspond to rosette centres. We find that the more cells that are present, increasing from four to seven cells and above, the more persistent the apical gaps are in *sdk* mutants (Fig 5F). Whereas in the wild type, apical gaps are more transient and rapidly resolved, in *sdk* mutants, the apical gaps persist, sometimes for the whole duration of GBE. In the latter case, we find that these then resolve when cell division starts in the epithelium at the end of GBE (S6B Fig). Based on these results, we conclude that the presence of Sdk facilitates the resolution of cortical discontinuities at the level of AJs during cell rearrangements in an extending tissue, this requirement being more acute when cells are rearranging as rosettes (involving more than four cells).

## Sdk is required for normal polarised cell intercalation during axis extension

We have shown previously that the tissue deformation of GBE is caused by a combination of cell intercalation and cell shape changes and that cell shape changes can compensate for cell intercalation defects [41, 43, 50]. We measure the relative contributions of these two cell behaviours by considering each cell and a corona of neighbours to calculate the different strain rates [50] (see Materials and Methods) (Fig 6A). Briefly, the relative movement of cell centroids in small patches of tissue is used to calculate the tissue strain rates; within each patch, individual cell shapes are approximated to ellipses to measure the cell shape strain rate; finally, the difference between tissue strain rates and cell shape strain rates gives a continuous measure of the strain rate due to cell intercalation (Fig 6A). Strain rates are then projected along the AP embryonic axis to calculate the rate of deformation in the direction of tissue extension. We find that the rate of tissue extension along AP is decreased in *sdk* mutants (Fig 6B and S7A and S7D Fig). Moreover, we find a decrease in cell intercalation contributing to extension (Fig 6D and S7C and S7F Fig), which is compensated to some extent by cell shape changes (Fig 6C and S7B and S7E Fig). This suggests that the relative contributions of cell intercalation and cell shape change to total tissue extension are altered in *sdk* mutants. Supporting this, we find that the proportion of the cell intercalation strain rate contributing to AP extension is indeed lower in *sdk* compared to the wild type (S8A Fig).

The above measure of cell intercalation is a measure of the continuous movement of cells relative to each other. We wanted to confirm the cell intercalation defect using a discrete measure. For this, we detected the number of neighbour exchanges, called T1 swaps, occurring for any group of four cells in the tissue. In this method, a T1 swap is defined by a loss of neighbour caused by cell–cell contact shortening, followed by the growth of a new cell–cell contact and a gain of neighbour (Fig 6E and S8B and S8C Fig; Materials and Methods) [49, 51]. While the total number of T1 swaps is only moderately decreased in *sdk* mutants compared to the wild type (S8D and S8E Fig), their orientation is abnormal. First, we find that in *sdk* mutants, the T1 swaps are not as well-oriented relative to the embryonic axes compared to the wild type (Fig 6F and S8H Fig) (note, however, that the orientation of the shortening junctions relative to the growing junctions is unchanged in *sdk* mutants compared to the wild type; S8F and S8G Fig). Second, we can quantify the contribution of T1 swaps to AP extension (defined as 'productive' T1 swaps; see Materials and Methods), and those are robustly decreased in *sdk* versus

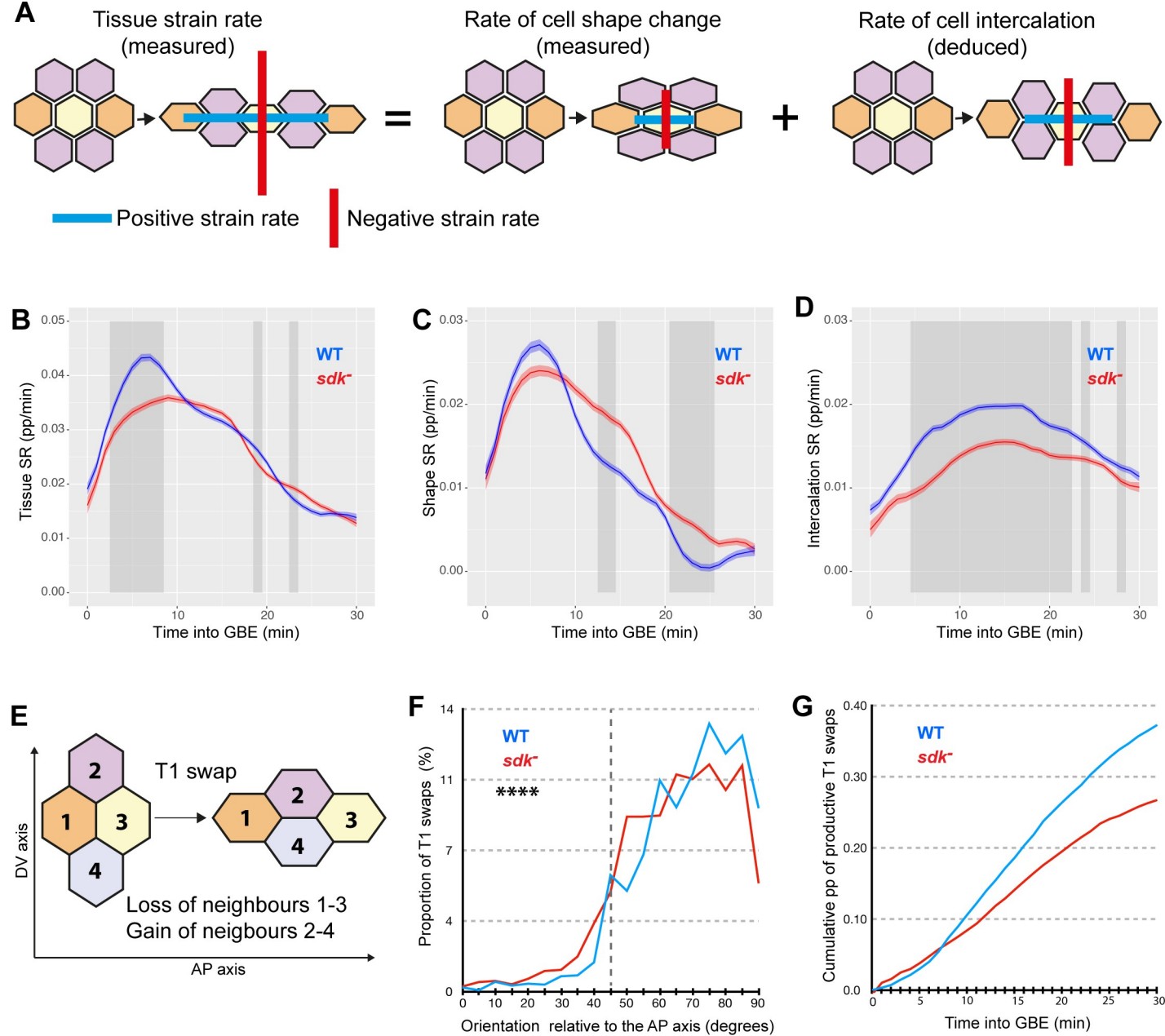

**Fig 6. *Sdk* is required for normal polarised cell intercalation.** (A) Graphical illustration of our measures of tissue and cell shape SRs (Materials and Methods). The cell intercalation SR is derived from these two measures. (B–D) Average SRs in the direction of extension (along AP) for five WT (blue) and five *sdk* mutant (red) embryos for the first 30 minutes of GBE. Total tissue SR (B), cell shape SR (C), and cell intercalation SR (D). Units are in pp per minute. (E) Diagram of a T1 transition leading to a loss of neighbours 1 and 3 along AP and a gain of neighbours 2 and 4 along DV. (F,G) Analysis of the number and orientation of T1 transitions averaged for five WT (blue) and five *sdk* mutant (red) embryos for the first 30 minutes of GBE (see also S8D and S8E Fig). (F) Orientation of all T1 transitions relative to the AP embryonic axis. Orientation is given by the angle of cell interfaces relative to AP, 5 minutes before a T1 swap (Kolmogorov–Smirnov test, N = 1,786 for WT and 1,890 for *sdk* mutant, D = 0.1115, p < 0.0001). (G) Cumulative proportion of T1 swaps contributing to axis extension in AP (called productive T1 swaps; see Materials and Methods) for the first 30 minutes of GBE and expressed as a pp of DV-oriented interfaces tracked at each time point. Data for graphs B–G can be found at https://doi.org/10.17863/CAM.44798. AP, anteroposterior; DV, dorsoventral; GBE, germband extension; pp, proportion; Sdk, Sidekick; SR, strain rate; WT, wild type.

the wild type (Fig 6G and S8I Fig). We also looked at the geometric arrangement of cells during junctional shortening. We find that the angle between the shortening junctions and the centroid–centroid line between intercalating cells is larger in *sdk* mutants compared to the

wild type (S8J Fig), suggesting that cell intercalation patterns are less regular. In conclusion, both the continuous and discrete methods we employed above indicate that the polarised cell intercalation contributing to AP tissue extension is decreased in *sdk* mutants.

## Modelling the *sdk* mutant phenotype

One hypothesis to explain a defect in polarised cell intercalation in *sdk* mutants is that the Sdk homophilic adhesion molecule facilitates the transition between shortening and elongating junctions at apical tricellular vertices. The Sdk adhesion molecule might provide a specialised adhesion system at vertices (perhaps bridging the intercellular vertex gap more effectively than the shorter E-Cadherin) or a specialised anchorage of the actomyosin cytoskeleton. To test this hypothesis, we extended our previously published vertex model of an intercalating tissue [49]. Vertex models traditionally implement cell rearrangement by imposing an instantaneous T1 swap on all small edges (below a threshold length) (Fig 7A). In order to model a putative phenotype in cell rearrangement, we developed a new framework in which the vertices of a shrinking edge temporarily merge to form higher-order vertices, which may resolve with some probability per unit time (Fig 7B and 7C and S1 Text). Vertices can be of rank 4 (four cells around a vertex; Fig 7B), to model a single T1 transition, or of rank 5 and above (five cells or more around a vertex; Fig 7C), to model rosettes. In addition to this change, we imposed periodic boundary conditions on the tissue, reducing artefacts that arise with a free boundary. Finally, we added a posterior pulling force to simulate the effects of the invaginating midgut [42, 43] (S9A Fig and S1 Text).

We used this new mathematical framework to model the cell intercalation defect we report for *sdk* mutants. First, we took into account the striking relationship between the number of cells involved in an intercalation event and the persistence of apical gaps or tears in *sdk* mutants (Fig 5F). Gaps forming between four cells, presumably as a consequence of a single T1 swap, take longer to close in *sdk* mutants compared to the wild type, but they eventually resolve. In contrast, gaps present at the centres of rosettes involving five cells or more often persist until the end of imaging (Fig 5F and S6B Fig). Second, our evidence indicates that a rosette centre is in fact made up of several separable Sdk string-like structures (Fig 3). Together, these results suggest that i) single T1 swaps might be delayed in *sdk* mutants and ii) this delay might increase when cells intercalate as rosettes because it requires the resolution of several T1 swaps in short succession. Our data in Fig 5F support the idea that rosettes accumulate and get stuck in *sdk* mutants (see also S1 and S2 Movies). To test whether intercalation might also be delayed in single T1s, we measured the resolution phase of T1 swaps in *sdk* mutants versus the wild type (S9B Fig). Note that in this analysis, only the successful T1 swaps are quantified, so this automatically excludes any stuck rosettes. We find a 1-minute delay between the wild type and *sdk* mutants (S9C Fig), supporting the assumptions of the model.

To model such a delay in *sdk* mutants, we imposed a lower probability of successful resolution of T1 swaps per unit time than in the wild type (Fig 7B and 7C). We distinguished isolated T1 swaps involving only four cells (Fig 7B) from linked T1 swaps in rosettes involving five or more cells by lowering the resolution probability further for rosettes (vertices connected to five or more cells) (Fig 7C). Rosettes appear in both *sdk* mutant and WT simulations, but as expected from the probabilities imposed, they get stuck in the *sdk* simulation, while they are resolved quickly in the wild-type simulation (Fig 7D, 7E and 7G) (S3 and S4 Movies). This leads to the topology of the cellular network being different in the *sdk* simulation in ways reminiscent of the *sdk* phenotype (see Fig 4A and 4B). We next compared the tissue strain rates in these simulations (Fig 7F and 7H). The strain rates are initially very similar, suggesting that the posterior pulling force is the main contribution to the initial AP strain rate. Then, the imposed

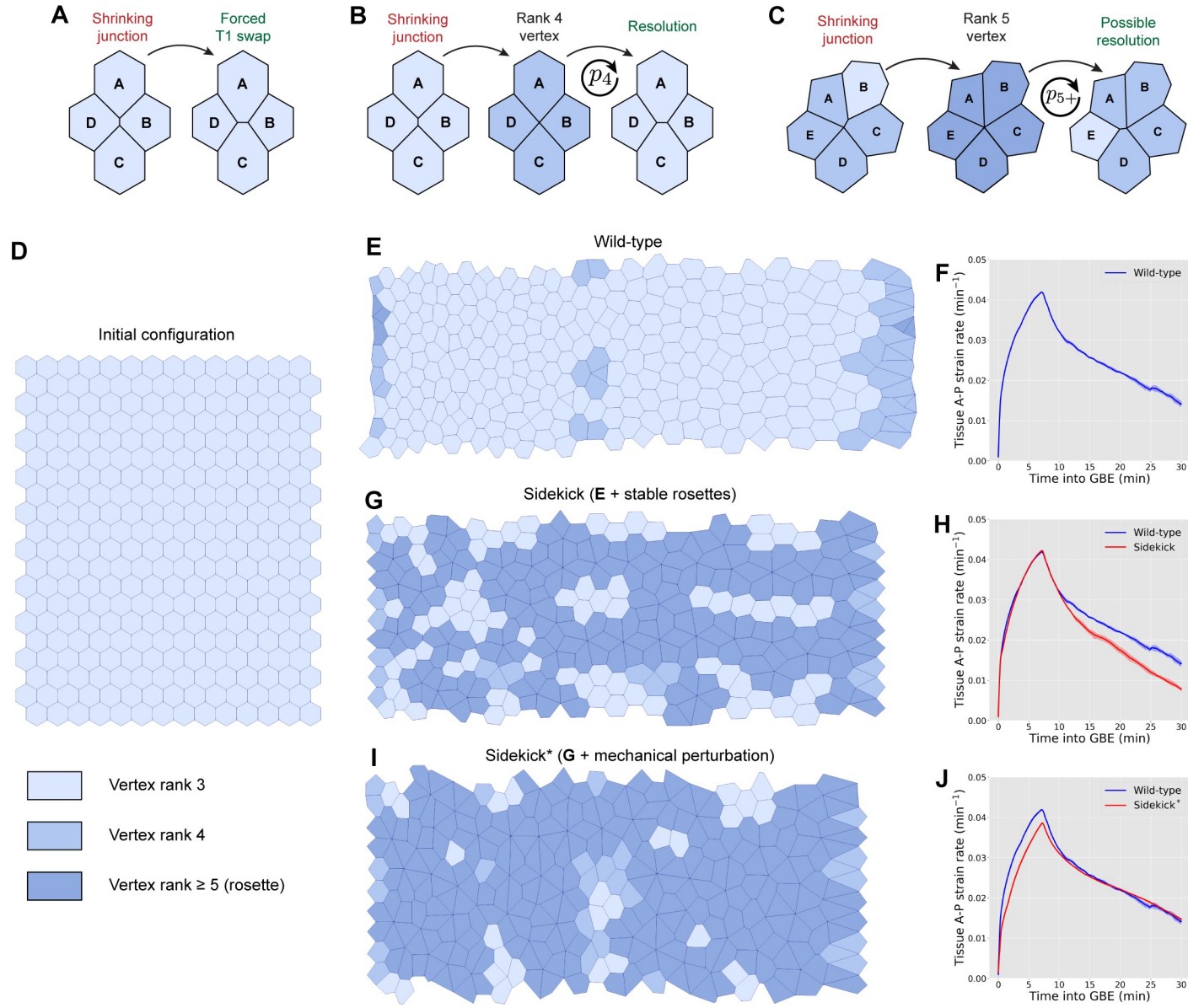

**Fig 7. Vertex models of the *sdk* mutant phenotype.** (A) Cell rearrangement (T1 transition) is usually implemented in vertex models as follows: edges with length below a threshold are removed, and a new edge is created between previously non-neighbouring cells. (B–C) Alternative implementation of cell rearrangement used in this paper. (B) Shortening junctions merge to form a four-way vertex and a protorosette, which has a probability, $p_4$, of resolving at every time step. (C) Formation of rosettes around higher-order vertices (formed of five cells, as shown here, or more) due to the shortening of junctions connected to four-way vertices. Edges connected to the shortening junction are merged into the existing vertex, which now has a probability, $p_{5+}$, of resolving at every time step. (D) Initial configuration for each simulation. The tissue is a tiling of $14 \times 20$ regular hexagons with periodic boundary conditions. All cells are bestowed one of four stripe identities, $\{S_1, S_2, S_2, S_4\}$, representing identities within parasegments, as in [49] (see S9A Fig for an illustration). (E) Wild-type simulation of *Drosophila* GBE in the presence of a posterior pulling force, implementing cell rearrangements as outlined in B and C with $p_4^{WT} = 1/\text{min}$ and $p_{5+}^{WT} = 0.1/\text{min}$. Model parameters used were $(\Lambda, \Gamma) = (0.05, 0.04)$. (F) Tissue strain rate in the A–P (extension) direction for wild-type tissues with parameters used in E. Solid line and shading represent mean and 95% confidence intervals from five independent simulations. (G) Simulation of GBE in a tissue in which T1 swaps are less likely to resolve, with $p_4^{sdk} = 0.1/\text{min}$ and $p_{5+}^{sdk} = 0/\text{min}$. All other parameters are kept equivalent to wild-type simulation. (H) Tissue strain rate in the A–P (extension) direction for tissues with parameters used in G compared to strain rate of wild-type tissue in E. Solid line and shading represent mean and 95% confidence intervals from five independent simulations. (I) Simulation of GBE in a tissue in which T1 swaps are less likely to resolve, with $p_4^{sdk} = 0.1/\text{min}$ and $p_{5+}^{sdk} = 0/\text{min}$, as in G, and additionally in which the shear modulus of the tissue (in the absence of actomyosin cables) has been reduced by setting $\Gamma = 0.01$. All other parameters are kept equivalent to wild-type simulation. (J) Tissue strain rate in the A–P (extension) direction for tissues with parameters used in I compared to strain rate of wild-type tissue in E. Solid line and shading represent mean and 95% confidence intervals from five independent simulations. As shown in key, for B, C, E, G, and I, cell colouring indicates the vertex rank of a cell, defined as the maximum number of cells sharing one of its vertices (note that darker blue is for rosettes of rank 5 and above). Further details about models and simulations can be found in S1 Text. Data for graphs F, H, and J can be found at https://doi.org/10.17863/CAM.44798. A–P, anterior–posterior; GBE, germband extension; Sdk, Sidekick.

delay in rearrangements leads to a reduced strain rate as the strength of the pull declines and DV-oriented junctions have shortened to the point of rearrangement (Fig 7H). When compared with the biological data, however, these strain rate patterns did not mimic the initial decrease in tissue strain rate in *sdk* mutants compared to the wild type (Fig 6B).

To capture the difference in the initial tissue-level strain rate, we made one additional modification to the model. We reasoned that the loss of Sdk may lower intercellular adhesion globally in the tissue since vertebrate Sdk homologues are known homophilic adhesion molecules [27, 28]. To model this, we perturbed the mechanical properties of the *sdk* tissue (in a manner equivalent to a decrease in the shear modulus of a mechanically homogenous tissue; see S1 Text). The posterior pull and the properties of the neighbouring tissues were left unchanged. In the new *sdk* simulation, the initial extension strain rate is now lower compared to the wild type, more accurately reproducing the biological data (Fig 7J; compare with Fig 6B). Interestingly, more rosettes form in this simulation (Fig 7I and S5 Movie). Because of the probability imposed for rosette resolution, this results in more stuck rosettes in this simulation, which, in turn, has an increased impact on the topology of the cellular network (Fig 7I). In conclusion, our mathematical modelling supports the notion that together, a change in the elastic mechanical properties of the cells and a delay in cell rearrangement could explain the *sdk* mutant phenotypes we observe in vivo.

## Discussion

In this paper, we identify the adhesion molecule Sdk as a resident protein of tricellular contacts at the level of AJs (tAJs) in *Drosophila*. To our knowledge, this is the only protein found specifically at this location in either invertebrates or vertebrates. Indeed, in *Drosophila*, other AJ proteins such as Canoe and many actin-binding proteins are enriched at tricellular contacts but also present at bicellular contacts [19–22]. Sdk, in contrast, is present specifically at tAJs in the large majority of *Drosophila* epithelia we surveyed. Its presence at the level of AJs is also distinct from the localisation of proteins marking tricellular occluding junctions, namely Gliotactin, Anakonda, and M6 at *Drosophila* tSJs and tricellulin and angulins at vertebrate tTJs [1, 6].

We show here that the localisation of Sdk does not require the contribution of three cells at tricellular contacts and that two cells contributing Sdk are sufficient. This suggests that Sdk molecules form homophilic adhesions between cell pairs at tricellular vertices. Consistent with this finding, biochemical and structural data support the notion that the vertebrate homologues of Sdk are homophilic adhesion molecules [24, 29]. This binding of Sdk in homophilic pairs is in contrast with the *Drosophila* protein Anakonda or the vertebrate protein angulin-2, which is thought to form tripartite complexes at tricellular occluding junctions [8, 52]. If a contribution of Sdk from three cells is not required, then this raises the question of the mechanism by which Sdk ends up at tricellular contacts in epithelia. Broadly, two hypotheses can be considered. One possibility is that an unknown molecular pathway targets Sdk to tricellular contacts. To target tricellular rather than bicellular contacts, such a pathway would need to include proteins that recognise special features of tricellular membranes such as curvature or a specialised actin cortex. In neurons, vertebrate Sdk proteins bind via their intracellular PDZ domain binding motif to members of the Membrane-associated guanylate kinase (MAGUK) family scaffolding proteins MAGI [27, 28]. Based on this, the *Drosophila* homologue Magi would be a candidate for binding the intracellular domain of *Drosophila* Sdk [53]. It remains to be seen whether such interaction could explain the tricellular localisation of Sdk. A different class of hypotheses is raised by the length of Sdk, which is more than three times the length of E-Cadherin. Because of their geometry, vertices in epithelia might have larger intercellular spaces than bicellular contacts. A possibility, therefore, is that Sdk resides at tricellular contacts

because of a sizing mechanism that excludes Sdk from bicellular spaces and concentrates it at tricellular spaces. For example, evidence for a sizing mechanism has been shown in vitro for bicellular contacts, whereby engineered proteins with different length in the intercellular space can sort from each other [54]. Further work will aim to distinguish between these hypotheses.

It is worth noting that although Sdk is normally at tricellular contacts in *Drosophila* epithelia, we have found that in the follicular cells after stage 7 and in larval salivary glands, Sdk is at the bicellular membranes. As hypothesised above, specific components targeting Sdk to tAJs might be missing, or the physical configuration of the bicellular space might be different in those cells. We also observed a change in Sdk localisation at shortening junctions during polarised cell intercalation. One possibility is that Sdk remains at tricellular contacts, and what we observed is in fact a thin membrane protrusion following the shortening contact (S1H Fig). This is compatible with our observation that the Sdk 'strings' extending from vertices into shortening junctions are never discontinuous. Alternatively, Sidekick might be invading the bicellular contact at shortening junctions (S1I Fig). We have tried with structured illumination to distinguish between a tricellular versus bicellular localisation of Sdk at shortening junctions. The increase in resolution was not sufficient to draw a conclusion, and better super-resolution techniques will be required to distinguish between the two possibilities proposed in S1H and S1I Fig.

Because of its unique localisation at the level of AJs at tricellular contacts, our starting hypothesis for the function of Sdk in epithelia was that it could be important in tissues where AJs are actively remodelled. Our findings support this hypothesis. We find that tissue remodelling does not occur normally during GBE in *sdk* mutants. Our quantifications demonstrate that on the apical side of the cells, cell shape changes, cell cortex organisation (at vertices), and polarised cell intercalation are all abnormal in *sdk* mutants compared to the wild type. Our mathematical model supports the notion that a delay in cell rearrangements in *sdk* mutants contributes to these defects. Sdk might facilitate T1 swaps through homophilic adhesion and/ or anchorage to the actomyosin cytoskeleton at tricellular vertices. Alternatively, Sdk may be affecting the mechanical properties of the tissue, reducing the speed at which junctions can shrink and/or grow (see below). In support of a delay in cell rearrangement, we find a measurable difference in the resolution of successful T1 swaps in *sdk* mutants (S9C Fig). In addition, we find that apical gaps or tears in the cortex form and persist at vertices contributed by four cells and above. The prevalence and persistence of these apical gaps is particularly acute at rosette centres formed by five cells or more. An explanation for this could be that Sdk is important for maintaining tricellular vertex integrity during rosette formation, allowing the resolution of successive T1 swaps. This is supported by our super-resolution data, which show that in the wild type, separable Sdk-YFP strings are always found at the centres of rosettes. It is possible that in *sdk* mutants, this partitioning does not occur and that tricellular vertices coalesce together during rosette formation. These multicellular vertices might be unable to undergo successive T1 swaps, leading to rosette resolution failure for the remainder of GBE, as was suggested by our data in Fig 5F.

Our analyses of *sdk* mutants also give insight into the dynamics of cell intercalation at the tissue-scale. It is striking that in our first mathematical simulation, a delay in T1 swaps has little effect on the initial phase of tissue-extension strain rate. However, the cell shapes become abnormal due to the increased presence of rosettes. This highlights how the balance between the rate of polarised intercalation and the mechanical properties of the tissue is important for maintaining isotropic cell shapes. In addition to introducing a delay in T1 swaps, it is possible that the absence of Sdk changes the general mechanical properties of the tissue, for example, via decreasing intercellular adhesion. Our second simulation attempts to capture this, mimicking better the initial decrease in the extension strain rate we find in *sdk* mutants. More rosettes

appear in this simulation, and as a consequence, the topology of the cellular network becomes more abnormal. This supports the notion that the in vivo phenotype of *sdk* mutants might be the consequence of both a delay in T1 resolution (worse in rosettes) and a change in the mechanical properties of the converging and extending tissue. Future work will aim at characterising the contribution of Sdk to the mechanical properties in this system.

Mutations in *sdk* are viable [23], suggesting that compensatory pathways exist for any epithelial remodelling mechanisms involving the Sdk adhesion molecule. In GBE, a pathway has been recently identified that promotes the intercalation of cells on their basolateral sides through protrusive activity [55]. While they both act downstream of AP patterning, apical-junctional and basolateral pathways are thought to be independent. This suggests that in *sdk* mutants, the basolateral protrusive activity might be intact and could rescue, in part, polarised cell intercalation, explaining the moderate defects we observe. The apical phenotypes (gaps in the cortex, abnormal cell shapes, and reduced intercalation) themselves are short-lived in *sdk* mutants, and we find that cell rearrangements caused by cell divisions can resolve these defects (S6B Fig). Since each cell in the ectoderm divides at least twice after the end of GBE, this is likely to account for restoring isotropic apical shapes to cells in the embryonic epithelium. Despite the viability of *sdk* mutants, two studies published while this article was under review report cell intercalation defects in *sdk* mutants in three other *Drosophila* tissues: the embryonic trachea and the pupal retina [56] and the genitalia disc [57]. This indicates that a role of Sdk in cell rearrangements might be widespread in *Drosophila* epithelia.

## Materials and methods

### Fly strains

We used the null alleles *sdk*$^{MB5054}$ (caused by the insertion of the Minos transposable element) and sdk$^{\Delta15}$ (a small deletion) [26]. Note that null *sdk* alleles are viable, and the flies could be kept homozygous/hemizygous (*sdk* is located on the X). We also used the null allele *sqh*$^{AX3}$ (*sqh* encodes the Myosin II regulatory light chain) in combination with *sqh-FP* constructs to label Myosin II as described in [58]. The Cambridge Protein Trap Insertions lines CPTI-000337, CPTI-000812, and CPTI-001692 all tag endogenous *sdk* with YFP via the insertion of a PiggyBac transposable element [44] (S1A–S1C Fig). Other transgenes used were Gap43-m-Cherry [59] to label cell membranes; ubi-DE-Cad-GFP [48], DE-Cad-GFP$^{KI}$ [60], and DE-Cad-mCherry$^{KI}$ [60] to label AJs; sqh-GFP$^{42}$ [58] and sqh-mCherry [61] to label Myosin II; and hs-flp$^{38}$ to induce clones [46].

### Genotypes

The genotypes used for main figures were as follows. Fig 1: (B–E, G) sdk-YFP$^{CPTI-000337}$, (F) sdk-YFP$^{CPTI-000337}$;Gap43-mCherry/CyO, and (H) FRT19A *sdk*$^{\Delta15}$/FRT19A *sdk*$^{\Delta15}$ clones surrounded by FRT19A *sdk*$^{\Delta15}$/FRT19A nls-RFP sdk-YFP$^{CPTI-000337}$ tissue. Fig 2: (A) sdk-YFP $^{CPTI-000337}$;DE-Cad-mCherry$^{KI}$. (C–E) sdk-YFP$^{CPTI-000337}$. Fig 3: (A–C) sdk-YFP$^{CPTI-000337}$;Gap43-mCherry/CyO and (B) sdk-YFP$^{CPTI-000337}$. Fig 4:; ubi-DE-Cad-GFP and *sdk*$^{MB5054}$;ubi-DE-Cad-GFP. Fig 5: (A–B) *sqh*$^{AX3}$, *sdk*$^{MB5054}$;DE-Cad-GFP$^{KI}$, sqh-mCherry; (C) *sqh*$^{AX3}$, *sdk*$^{MB5054}$;sqh-GFP; (D–F) *sqh*$^{AX3}$;DE-Cad-GFP$^{KI}$, sqh-mCherry and *sqh*$^{AX3}$, *sdk*$^{MB5054}$;DE-Cad-GFP$^{KI}$, sqh-mCherry. Fig 6: ubi-DE-Cad-GFP and *sdk*$^{MB5054}$; ubi-DE-Cad-GFP.

The genotypes used for supporting figures are as follows. S1 Fig: (C) sdk-YFP$^{CPTI-000337}$; sdk-YFP$^{CPTI-000812}$;sdk-YFP$^{CPTI-001692}$ and (D–G) sdk-YFP$^{CPTI-000337}$. S2 Fig: sdk-YFP$^{CPTI-000337}$. S3 Fig: (A,B) sdk-YFP$^{CPTI-000337}$. S4 Fig: (A,C) ubi-DE-Cad-GFP. (B,D) *sdk*$^{MB5054}$;ubi-DE-Cad-GFP. S5 Fig: ubi-DE-Cad-GFP and *sdk*$^{MB5054}$;ubi-DE-Cad-GFP. S6 Fig: *sqh*$^{AX3}$, *sdk*$^{MB5054}$;

DE-Cad-GFP$^{KI}$, sqh-mCherry. S7, S8, and S9B and S9C Figs: ubi-DE-Cad-GFP and $sdk^{MB5054}$; ubi-DE-Cad-GFP.

## Clonal induction

$sdk^{\Delta15}$ mutant clones were induced in the follicular epithelium in the ovaries using the FRT/FLP system [46]. L3 larvae from the cross FRT19A nls-RFP, sdk-YFP;hs-flp$^{38}$ × FRT19A, $sdk^{\Delta15}$ were heat-shocked at 37˚C for 2 h every 12 h until pupariation. Ovaries from female progeny were dissected for immunostaining.

## Immunostainings

Embryos were collected on apple plates, aged to the desired stage, then dechorionated in 100% commercial bleach for 1 minute and rinsed in tap water. Embryos were fixed at the interface between heptane and 37% formaldehyde for 5 minutes. Embryos were then washed in PBS and PBS with 0.1% Triton-100 and devitelinised by hand. Embryos were washed twice in PBS with 0.1% Triton-100, followed by a 30-minute blocking incubation in PBS + 1% BSA. Embryos were incubated overnight with primary antibodies at 4˚C and washed 3 times for 10 minutes in PBST. Embryos were incubated with secondary antibodies for 1 h at room temperature and then washed 3 times for 10 minutes in PBS with 0.1% Triton-100 before mounting in Vectashield containing DAPI (Vectorlabs Catalog No. H-1000; Burlingame, CA, USA) (with the exception of super-resolution imaging; see below). For adult midgut and ovaries, tissue was dissected from 2-day–old females and heat-fixed for 30 seconds at 100˚C. Fixed tissue was incubated overnight with the primary antibody, followed by secondary antibodies at 4˚C with 3 × 10 minutes washes in PBST following both incubations.

Primary antibodies and their dilutions were rat anti-DE-Cad 1:300 (Developmental Studies Hybridoma Bank #DCAD2; University of Iowa, Iowa City, IA, USA), chicken anti-GFP 1:200 (Abcam #ab13970; Cambridge, UK), goat anti-GFP conjugated with FITC 1:200 (Abcam #ab6662), mouse anti-Dlg 1:100 (DSHB #4F3), rabbit anti-RFP conjugated with CF594 1:1,000 (Biotium #20422; Fremont, CA, USA), rabbit anti-aPKC 1:200 (Santa Cruz Biotechnology sc216; Dallas, TX, USA), mouse anti-Sdk 1:200 [26], and mouse anti-Arm 1:100 (DSHB #N2 7A1). Cell membranes were stained using the lectin fluorescent conjugate Concanavalin A, Alexa Fluor 594 Conjugate (Thermo Fisher Scientific C11253; Waltham, MA, USA) at 1:1,000 dilution.

Secondary antibodies (diluted 1:500) were sourced from Abcam or Life Technologies (Carlsbad, CA, USA). F-Actin was stained with phalloidin conjugated with CF594 or CF568 (1:500) (Biotium #00044 and #00045, respectively).

## Confocal imaging of fixed tissues

Immunostained tissues were imaged on a Nikon D-Eclipse C1 TE2000-E scanning confocal with a Nikon 40× PlanApo (NA 1.3; Nikon, Tokyo, Japan) or a Nikon 60× PlanApo (NA 1.4) oil objectives or on a Leica SP8 with Leica HC PL 40× (NA 1.3; Leica, Wetzlar, Germany) or Leica HC PL 63× (NA 1.4) oil objectives. Immunostained ovaries were imaged on a Leica SP5 confocal system using a Leica HCX PL Apo CS Oil 63× (NA 1.4) objective.

## SIM imaging and analysis

Immunostained embryos were mounted in SlowFade Diamond Antifade mountant (Molecular Probes, refractive index 1.42; Eugene, OR, USA) after application of a glycerol series up to 75% glycerol in PBS under a coverslip to match the refractive index of the imaging system. Images were acquired using an OMX microscope (Applied Precision, Issaquah, WA, USA) in

super-resolution mode, with an Olympus PlanApoN Oil 60× oil immersion lens (NA 1.42; Olympus, Tokyo, Japan) and 1.515 refractive index immersion oil (Applied Precision). Embryos were first mapped with a DeltaVision inverted widefield microscope with the stage mapped to the OMX stage. Z-stacks were imaged at 0.125-μm intervals and widefield image deconvolution, and super-resolution reconstruction was done using SoftWoRx software (Applied Precision). Image acquisition and SIM reconstruction parameters were modified based on the quality and type of stainings, guided by analysis from the FIJI Toolbox SIMcheck [62]. Channel-specific Wiener filters based on the filter optimum suggested by this software were used with values >0.004 to prevent smoothening of detail because the software did not optimise for the nature of signal present in the acquired images (sparse, bright signal).

The path of Sdk-YFP strings (Fig 2C–2H, Fig 3B and S3 Fig) were followed manually and measured in three dimensions through reconstructed super-resolution images using the FIJI plugin 'Simple neurite tracer' [63].

## Confocal imaging of live embryos

After dechorionation in bleach and rinsing well in tap water, embryos were mounted in Voltalef 10S oil (Arkema, Pierre Benite, France) on a custom-made mount between a coverslip and a Lumox $O_2$-permeable membrane (Sarstedt, Newton, NC, USA).

Images presented in Fig 2A, Fig 5B, S1C Fig and S6 Fig were acquired using a Nikon Eclipse E1000 microscope coupled to a Yokogawa CSU10 spinning disc head (Tokyo, Japan) and Hamamatsu EM-CCD camera (Hamamatsu City, Japan) controlled by Volocity software (Perkin Elmer, Waltham, MA, USA) with a Nikon 60× PlanApo oil objective (NA 1.3). Images in Fig 3A and Fig 5A were acquired on a Leica SP8 scanning confocal with a Leica HC PL 63× (NA 1.4) oil objective.

For cell tracking and cell behaviour analysis (Fig 4, Fig 6, S4 Fig, S5 Fig, S7 Fig and S8 Fig), movies were acquired as described in [49]. Briefly, embryos were imaged under a 40× NA 1.3 oil objective on a spinning disc confocal, consisting of a Nikon Eclipse E1000 microscope coupled to a Yokogawa CSU10 spinning disc head and Hamamatsu EM-CCD camera controlled by Volocity software (Perkin Elmer). All movies for tracking were recorded at 21°C ± 1°C for consistency in developmental timing. Embryos were imaged ventrally, and z-stacks were acquired at 1-μm intervals every 30 seconds from stage 6 onwards. The viability of the embryos was checked postimaging by transferring the imaging apparatus to a humid box at 25°C. In the rare eventuality that embryos did not hatch, the corresponding movies were discarded.

## Analysis of apical epithelial gaps

We use the terms apical epithelial gaps or tears to refer to discontinuities in the usual apposed localisation of actomyosin and E-Cadherin along cell–cell contacts where the membranes of two or more cells meet. As observed in our movies, apical gaps are first detected by a decrease in local E-Cadherin signal, suggesting that local adhesion between cells has been reduced. We assume that cell membranes at the level of AJs become separated, but we have not been able to image single membranes convincingly. By contrast, Myosin II enrichments surrounding these apical gaps are clearly observable (S6A Fig). We therefore defined apical gaps in our movies as an observable ring in the Myosin II channel that arises between previously abutting cells and that is subsequently eliminated. These gaps are distinct from the known delamination events occurring in GBE.

The presence of gaps was quantified in movies of wild-type and *sdk* mutant embryos labelled with sqh-GFP-mCherry and DE-Cad-GFP[KI] (see genotype list), acquired on a confocal spinning disc at 40× magnification (see above) in order to view a large area of the germband. Observations were made on z-stack projections of up to 40 minutes of GBE. We

manually followed Myosin II rings during GBE in order to quantify the persistence of those gaps from their appearance to their disappearance (Fig 5E and 5F). For this analysis, we did not include data on gaps that did not resolve within the duration of the movie (over 40 minutes). The number of gaps in view for each embryo movie (Fig 5D) was quantified at the image frame when the first ventral midline cell divided (with a dumbbell shape), which corresponds to 26–30 minutes into GBE in WT [41]. The number of gaps was then expressed relative to the area of the embryo in view.

## Image segmentation and cell tracking

Image segmentation and the tracking of cell centroids and cell–cell interfaces was performed using the custom-made software 'otracks' as in [41, 43, 49]. Briefly, confocal z-stacks of movies of genotype ubi-DE-Cad-GFP or $sdk^{MB5054}$;ubi-DE-Cad-GFP were used to identify an apical plane at the level of AJs that follows the curvature of the embryo for image segmentation. Note that we used ubi-DE-Cad-GFP rather than DE-Cad-GFP$^{KI}$ because the signal was brighter with ubi-DE-Cad-GFP, and, as a consequence, the segmentation of the cell contours was easier. The tracking software identifies cells and links them through time using an adaptive watershedding algorithm. Following automated tracking, manual correction of the tracks was performed for each of the five wild-type and five $sdk$ mutant movies analysed in this study. An example of a tracked movie for each genotype is provided in S1 and S2 Movies. For each cell at each time point, coordinates of cell centroids, perimeter shapes, cell–cell interfaces, and links forwards and backwards in time for both cells and interfaces are stored.

## Movie synchronisation, cell type selection, and embryonic axes orientation

For each genotype, movies were synchronised at the time point when the tissue strain rate in the AP axis exceeded 0.01 proportion per minute. This movie frame was set to time 0 of GBE (see S4A and S4B Fig). This time point corresponds to the end of mesoderm invagination (see S1 and S2 Movies).

Note that for calculating strain rates and all further analyses, we included only neurectoderm cells (the population of cells that undergo convergent extension), having classified and excluded all head, mesoderm, mesectoderm, non-neural ectoderm, and amnioserosa cells, as previously described in [49]. The resulting cell population tracked is shown with purple centroids in the example S1 and S2 Movies. For each genotype, the number of cells tracked and selected for analysis at each time point is shown in S4C and S4D Fig. Although the movies are longer, in this study, we focused our analyses on the first 30 minutes of GBE from the zero defined during movie synchronisation.

To be able to measure angles relative to embryonic axes in some of the analyses, the orientation of the ventral midline at the start of GBE was used as the orientation of the AP axis. Note that some embryos rolled a little in the DV axis; this is taken into account and corrected.

## Cell shape analyses

Cell shapes are approximated by best-fit ellipses that are constrained to have the same area and centroid as the raw pixelated cell. The best-fit ellipses are found by minimising the area of mismatch between pixelated contours and ellipses.

**Axial shape elongation (Fig 4C and 4D).** We formulated a single measure that encapsulated both the degree of cell shape anisotropy and the orientation of the cell's longest axis. Cell shape anisotropy is calculated as the log ratio of the fitted ellipse's major axis to the minor axis, giving values ranging from 0 (isotropic) up to around 1.5 (strongly elongated) in our data. The orientation of the fitted ellipse's major axis is measured relative to the embryonic axes (given

by the ventral midline orientation; see above), with 0˚ aligned with the AP axis and 90˚ aligned with the DV axis. We then calculated

$$\text{Axial shape elongation} = (\text{orientation}/45 - 1) * (\text{elongation log ratio})$$

This gives a range of values from around −1.5 to 1.5, in which negative values correspond to AP-elongated cells and positive values correspond to DV-elongated cells (see Fig 4C). Note that isotropic cells and cells oriented at 45˚ relative to the embryonic axes will score zero, irrespective of how much the latter cells are elongated.

**AP and DV cell lengths (Fig 4E and 4F).** To calculate the length of the cell along embryonic axes, we projected the cell shape ellipse onto the embryonic axis orientations, giving the diameter of the ellipse in those orientations.

**AP and DV interface lengths (Fig 4G and 4H).** The lengths of cell–cell interfaces are calculated as the straight-line length between neighbouring vertices (vertex–vertex line) along which three cells meet. We classified interfaces as being either AP- or DV-oriented, depending on whether the vertex–vertex line was less than or greater than 45˚ from the orientation of the AP axis, respectively.

**Contoured heat maps (S5 Fig).** Heat maps show the above measures as a function of time into GBE along the y-axis, plotted against DV location along the x-axis, with the heat scale representing the third variable. The third variable is averaged over the AP axis. Heat maps show the mean values of the third variable for each grid square of the plot, the population size of which is shown in 'N' heat maps. For example, for S5B Fig, the 'N' heat map is in S5C Fig, with 30 one-minute time bins in the y-axis and 20 DV-coordinate bins of 3 μm width in the x-axis.

## Strain rates analysis

Strain rates were calculated as previously [19, 41, 49]. Briefly, following cell tracking (see above), local tissue strain rates are calculated for small spatiotemporal domains, using the relative movement of tracked cell centroids [50]. For the analysis of GBE, we use small spatiotemporal domains composed of a focal cell and one corona of neighbouring cells over a 2-minute interval (contained within 5 movie frames) (Fig 6A). Such domains, focused on each cell in each movie frame, are located at cell apices, following the DV curvature and more gentle AP curvature of the germband surface. The domains are therefore first untilted and uncurved to give flat (2D) domain data. From these domains, two strain rates describing an ellipse are calculated, one in the orientation of greatest absolute strain rate and the other perpendicular to the first one [50]. These strain rates are then projected onto the embryonic axes (see above for the determination of embryonic axes) to find the sign and magnitude of the rate of tissue deformation along AP or DV. Graphs of average tissue strain rate over time for deformation along AP and deformation along DV are shown in S4A and S4B Fig.

Next, the tissue strain rates are decomposed into the additive contribution of two cell behaviours, cell shape change and cell intercalation [50] (Fig 6A). For each cell in the small spatiotemporal domains defined above, the rate of cell shape change is calculated using minimisation, finding the 2D strain rates that most accurately map a cell's pixelated shape to its shape in the subsequent time point. The area-weighted average of these individual cell shape strain rates is then taken for each domain. Intercalation strain rates, which capture the continuous process of cells in a domain sliding past each other in a particular orientation, are then calculated as the difference between the total tissue strain rates and the cell shape strain rates [50].

Strain rates were calculated using custom software written in IDL (code provided in [50]). Strain rates in units of proportional size change per minute can be averaged in space (along

AP or DV) or accumulated over time. In Fig 6 and S7 Fig, we average strain rates between five movies for each genotype and show within-embryo confidence intervals. For statistical tests, we employed mixed-effects models (see Statistics section below).

We also calculated the proportion of tissue extension accounted for by intercalation in wild-type and *sdk* mutant embryos for each local domain (S8A Fig). For these domains, we calculated the ratio of AP intercalation strain rate/AP tissue strain rate. We took the natural log of this ratio because log-ratio space is more appropriately linear for averaging and comparisons.

### Neighbour exchange analysis

**Detection of T1 swaps.** As previously, we used changes in neighbour connectivity of quartets of cells in our tracked cell data to identify neighbour exchange events (T1 processes) [49, 51] (Fig 6E–6G and S8 Fig). For many T1 swaps, we found that interface shortening to the swap was followed immediately by interface lengthening. In these cases, the consecutive image frames in which one pair of the quartet of cells handed over the interface to the other pair was straightforward to identify. However, there were also examples in which the precise timing of the T1 was less obvious, either for biological reasons or because of the vagaries of pixel ownership during image segmentation. We needed algorithmic rules to locate the timing of T1 events when topology was unresolved for periods of time, for example, when a quartet of cells met temporarily (for several frames) at a four-way vertex or when a quartet was involved in repeated back and forth topological swaps with very short interfaces. We therefore took the approach of making interfaces 'immortal', with the identity of a shortening interface transferred to the lengthening interface after a swap, forcing three-way connectivity through all movie frames of a T1 swap and smoothing out rapidly swapping interfaces. For four-way vertices, this meant assigning connectivity to one of the opposed cell pairs of the quartet even though no physical interface existed. This was normally the pair of the quartet that most recently shared an interface, though it also depended on the following rules for simplifying rapidly swapping T1s. When a quartet of cells swapped ownership of the included interface for periods shorter than 5 image frames (less than 2.5 min), ownership of the interface was retained by the cell pair that was connected before and after this short interlude. For multiple short bouts of swapping connectivity, interface ownership for the shortest bouts were reversed first. The above rules ensured that connectivity within a local quartet of cells was always known and that rapid changes of connectivity were smoothed over. After the above forced connectivity and swap smoothing was applied, 'T1 gain' events were defined in the frame when cells made first contact, and minutes before and after T1 swapping were calculated relative to this time origin. Note that this method does not distinguish between solitary T1s and T1s involved in rosette-like structures.

**Defining the resolution phase of a T1.** For an interface that has been detected as undergoing a T1 swap (using the above criteria), the resolution phase is defined as the time the interface spends with length less than 0.75 μm from before to after the interface swap (see S9B Fig). The 1D signal of interface length is filtered with a Hanning smoothing window to eliminate noise, during which the interface length may temporarily jump. Resolutions that are occurring when tracking begins or do not complete when tracking ends are not included.

**T1 swap counting and orientation.** The total number of T1 swaps are given in S8D and S8E Fig, normalised by the total number of tracked cell–cell interfaces. To analyse the orientation of T1 swaps, we measured either the orientation of the shortening interface before the T1 (shared by cells 1 and 3 in Fig 6E) or the lengthening interface after the T1 (shared by cells 2 and 4). Alternatively, we measured the orientation of the centroid–centroid line between the

pair of cells that gained connectivity through the T1 (line between centroid of cell 2 and centroid of cell 4 in Fig 6E). Orientations were measured with respect to the embryonic axes. For example, Fig 6F and S8H Fig show the orientation of the shortening interfaces 5 minutes before their T1 swap relative to the AP axis. The 5-minute interval was chosen as a compromise, ensuring that interfaces were both long enough to be measured accurately and also actively shortening (see, for example, S9B Fig).

For Fig 6G, we defined a measure of 'productive' T1 swaps, meaning swaps contributing to tissue extension [49, 51]. We first classified T1 swaps as either AP- or DV-oriented, depending on whether the centroid–centroid line between gaining neighbours was less than or greater than 45˚ from the AP axis, respectively. Not all T1 swaps were DV-oriented, with occasional AP-oriented swaps being subsequently reversed or, even more rarely, permanent. We therefore subtracted the AP-oriented gains from the DV-oriented gains to calculate the net number of productive T1 swaps contributing to tissue extension. In Fig 6G, to be able to compare genotypes, this measure was then normalised by the total number of DV-oriented interfaces.

**Other angular measures.** We compared the orientation of interfaces before and after T1 transition (S8F and S8G Fig). To do this, we first calculated the angular difference between the interfaces 5 minutes before and after the T1. We then controlled for rotation of the local four-cell domain by adding or subtracting any change in orientation of the centroid–centroid line of the gaining neighbours over this time period. For 'cartoon' T1s represented as hexagons (see Fig 6E), we expect this angle between shortening and elongating junctions to be 90˚. We find a distribution skewed towards 90˚ as expected (S8F Fig), with a mean around 75˚ all through the first 30 minutes of GBE (S8G Fig).

We further characterised the orientation of T1 swaps by comparing angles between interfaces and centroid–centroid lines. In S8J Fig, we measured the angle between the shortening interface in a T1 swap and the centroid–centroid line joining the cells that will gain contact during the swap. We monitored this angle for the first 15 minutes before T1 swap. For 'cartoon' T1s represented as hexagons (see Fig 6E), we expect this angle to be 0˚. In both the wild type and *sdk* mutants, this angle is between 12˚ and 17˚ (S8J Fig).

## Statistics

Average strain rate, cell shape, and junction length plots from tracked movies were generated in R, with profiles smoothed over 3 bins for presentation. Statistical significance was calculated on unsmoothed data. Statistical tests were performed using the 'lmer4' package and custom-written procedures in R as used previously in [41, 43, 51]. A mixed-effects model was used to test for significant differences between genotypes [64]. This test estimates the *p*-value associated with a fixed effect of differences between genotypes, allowing for random effects contributed by differences between embryos within genotypes. An independent test was performed at each time point in the analysis. Time periods during which the test shows a statistical difference of $p < 0.01$ are highlighted by dark grey on temporal plots.

For mixed-effects models, it is not possible to present a single overall confidence interval. Instead, we have a choice to show one of within- or between-genotype confidence intervals, and we have chosen the former, as previously [41, 43, 49, 51]. Therefore, error bars in time-lapse plots show an indicative confidence interval of the mean, calculated as the mean of within-embryo variances. The between-embryo variation is not depicted, even though both are accounted for in the mixed-effects tests.

## Mathematical model

The model is described in S1 Text.

## Supporting information

**S1 Movie. Example of tracked WT movie (embryo 2).** WT, wild type.
(MOV)

**S2 Movie. Example of tracked *sdk* movie (embryo 3).** Sdk, Sidekick.
(MOV)

**S3 Movie. Simulation of WT GBE, corresponding to Fig 7E.** GBE, germband extension; WT, wild type.
(MP4)

**S4 Movie. Simulation of *sdk* mutant GBE, corresponding to Fig 7G.** GBE, germband extension; Sdk, Sidekick.
(MP4)

**S5 Movie. Simulation of *sdk* mutant GBE, corresponding to Fig 7I.** GBE, germband extension; Sdk, Sidekick.
(MP4)

**S1 Text. Supplementary methods for the vertex model of *Drosophila* GBE in the wild type and *sdk* mutant.** GBE, germband extension; Sdk, Sidekick.
(DOCX)

**S1 Table. Summary of the localisation of Sdk-YFP in *Drosophila* epithelia.** Sdk, Sidekick; YFP, yellow fluorescent protein.
(DOCX)

**S1 Fig. Localisation of Sdk-YFP protein traps at tAJs.** (A and B) Schematics showing the genomic structure of the *sdk* gene (A) and the domains of the corresponding protein (B). Transposon insertions, alleles, and C-term location of the antibody epitope are indicated. C) All three YFP protein traps from the CPTI collection localise at vertices in the embryonic ectoderm, shown here in images of the ventral embryonic ectoderm in live embryos, taken by Claire Lye and Huw Naylor during our CPTI screen [19]. Scale bar = 20 μm. (D) Super-resolution SIM imaging of fixed embryos immunostained with Sdk-YFP and aPKC. Maximum projection (XY) and z-reconstruction (XZ). Scale bars = 1 μm. (E) Cartoon summarising the apicobasal localisation of Sdk in *Drosophila* epithelia based on SIM imaging in D. (F, G) Super-resolution SIM imaging of fixed embryos immunostained with Sdk-YFP and an antibody recognising a C-term epitope in Sdk [26]. (F) Maximum projection, apical view. Scale bars = 5 μm. (G) Close-ups of individual strings to show the colocalisation between Sdk-YFP and the Sdk antibody signal. Alignment between channels for super-resolution imaging was performed with the help of fluorescent beads. Scale bars = 1 μm. (H) In model 1, Sdk-YFP remains at tricellular contacts, and protrusions containing Sdk-YFP follow the shortening contact, explaining its apparent localisation at shortening junctions. (I) Alternatively, in model 2, Sdk-YFP molecules do not remain tricellular and invade the bicellular contact at shortening junctions. aPKC, Atypical protein kinase C; CPTI, Cambridge Protein Trap Insertion; Sdk, Sidekick; SIM, Structured Illumination Microscopy; tAJ, tricellular adherens junction; YFP, yellow fluorescent protein.
(TIF)

**S2 Fig. Localisation of Sdk-YFP in *Drosophila* epithelia.** Images show stainings or live imaging of Sdk-YFP in diverse epithelia from different developmental stages. (A) Hindgut, stage 13 embryo, fixed and immunostained tissue, maximum intensity projection. (B) Salivary glands,

stage 13 embryo, fixed and immunostained tissue, maximum intensity projection. (C) Eye imaginal disc posterior to the morphogenetic furrow. Dissected from third instar wandering larvae. Fixed and immunostained tissue, maximum intensity projection. (D) Salivary gland. Dissected from third instar wandering larvae. In this tissue, Sdk-YFP localises to all lateral and basal cell–cell junctions. Fixed and immunostained tissue, maximum intensity projection. (E) Follicular epithelium from stage 6 egg chamber from ovaries of adult female flies. Sdk-YFP localises to apical vertices at mitotic stages. Live imaging. Top: apical view, maximum intensity projection. Bottom: lateral view, single z-slice. (F) Posterior midgut of 3-day–old adult female flies. Fixed and immunostained tissue, lateral view, single z-slice. All scale bars = 20 μm. Sdk, Sidekick; YFP, yellow fluorescent protein.
(TIF)

**S3 Fig. Localisation of Sdk at rosette centres.** (A) Sdk-YFP string localisation at a rosette centre involving five cells, imaged by super-resolution SIM. The image is from a stage 8 embryo fixed and stained for GFP and the leptin Concanavalin A, a membrane binding protein. Maximum projection over 15 slices = 1.875 μm. Close-ups of the rosette centre with different projections are shown in yellow boxes to demonstrate that three distinct strings can be resolved in the apical-most projections. (B) Single z-slices of the stack shown in A at different apicobasal depths. Sdk-YFP strings represent the apical-most organisation of junctions. Yellow arrows point to junctions that have a different configuration in the z-slice 1.875 μm more basal. All scale bars = 2 μm (including in close-ups). GFP, green fluorescent protein; Sdk, Sidekick; SIM, Structured Illumination Microscopy; YFP, yellow fluorescent protein.
(TIF)

**S4 Fig. Movie synchronisation and cell counts.** (A–B) Summary of tissue deformation (strain) rates for five wild-type (A) and five *sdk* (B) embryos in the course of GBE. Tissue strain rates are plotted for both tissue extension along AP (full curves) and convergence along DV (dashed curves). All movies are synchronised to a time point corresponding to the extension strain rate first exceeding 0.01 (proportion per minute), which defines time 0 of GBE. In analyses throughout the paper, we summarise data for the first 30 minutes of GBE. Note that the positive deformation in DV (dotted curves) around the start of extension is due to the ectoderm tissue being pulled ventrally by mesoderm invagination. Averaged data between all five movies are shown as black curves for each genotype. (C,D) Numbers of cells tracked then selected for analysis for each wild-type and *sdk* movie (total cell number for each genotype in shown in black). The number of successfully tracked cells is low at the onset of GBE because fewer ventral ectodermal cells are in view because of mesoderm invagination. Data for graphs can be found at https://doi.org/10.17863/CAM.44798. AP, anteroposterior; DV, dorsoventral; GBE, germband extension; Sdk, Sidekick.
(TIF)

**S5 Fig. Quantification of cell shape changes in *sdk* mutants versus the WT.** (A) Coordinate system used for spatiotemporal plots shown in B–I. Spatial data are collapsed along AP and given as a function of location along the DV axis. Locations are indicated in μm from the ventral midline (0 at the midline, up to 70 μm laterally). Because of bilateral symmetry, we can mirror the data from the two halves of the embryo along the midline. This simplifies the DV coordinates, and we use x-axes showing locations from 10 to 70 μm. The y-axis gives the time from GBE onset. (B) Evolution of axial shape elongation (see also Fig 4C and 4D) for WT and *sdk* for the first 30 mins of GBE (y-axis) and as a function of cell position along DV (x-axis). (C) Number of analysed cells per bin for the same spatiotemporal parameters, for graphs B, D, E. (D,E) Spatiotemporal evolution of AP or DV cell length (see also Fig 4E and 4F). (F–I)

Spatiotemporal evolution of the lengths of AP or DV cell interfaces (see also Fig 4G and 4H). G and I give the number of AP or DV cell interfaces analysed for each spatiotemporal bin. Note that the raw data shown in all above panels are summarised in Fig 4D–4H. AP, antero-posterior; DV, dorsoventral; GBE, germband extension; Sdk, Sidekick; WT, wild type. (TIF)

**S6 Fig. Apical gap dynamics in *sdk* mutants.** (A) Representative example of the formation and resolution of an apical gap in an *sdk* mutant embryo labelled with DE-Cadherin and MyoII-Cherry. A projection of 3 μm (± 1 μm from AJ) is shown for each time point. (B) Representative example of a persistent apical gap in an *sdk* mutant embryo that is finally resolved when cells nearby the gap start dividing (stars marks dividing cells). AJ, adherens junction; MyoII, Myosin II; Sdk, Sidekick. (TIF)

**S7 Fig. Strain rates for all wild-type and *sdk* mutant embryos.** (A–C) Strain rates for each of five wild-type movies. (D–F) Strain rates for each of five *sdk* movies. (A,D) Total tissue strain rates. (B,E) Cell shape strain rates. (C,F) Cell intercalation strain rates. All movies are labelled with ubi-E-Cad-GFP (see Materials and Methods). Strain rates are along AP, the direction of tissue extension, and are given in pp per minute for the first 30 minutes of GBE. The average for each genotype is shown as a black curve. Data for graphs can be found at https://doi.org/10.17863/CAM.44798. AP, anteroposterior; E-Cad, E-Cadherin; GBE, germband extension; GFP, green fluorescent protein; pp, proportion; Sdk, Sidekick; ubi, Ubiquitin. (TIF)

**S8 Fig. Comparison of polarised cell intercalation in wild-type and *sdk* mutant embryos.** For all graphs, data shown are from the analysis of five wild-type and five *sdk* mutant embryos (see Materials and Methods). (A) Ratio of cell intercalation/tissue strain rates in AP in wild-type and *sdk* mutant embryos (see also Fig 6B–6D). (B–C) Detection of T1 swaps in tracked movies for a wild-type (B) and an s*dk* mutant embryo (C). Movie frames at 10 and 30 minutes into GBE show the cell interfaces that will be lost (blue) and gained (red), respectively, for the detected T1 swaps. (D–E) Cumulative curve of T1 swaps in any direction for the first 30 mins of GBE, expressed as a pp of all cell interfaces tracked at each time point. Average curves for wild-type and *sdk* embryos (D) and individual curves for each movie (E). (F) Angle between lost and gained cell interfaces during a T1 swap for the first 30 minutes of GBE. The orientation of cell interfaces is measured 5 minutes before and after a swap, respectively. (G) Same quantification as (F) but over the first 30 mins of GBE (x-axis). (H,I) Individual curves for each movie for the quantifications shown in Fig 6F and 6G, respectively. (J) Angle between the shortening cell interfaces in a T1 swap and the line between centroids of the future cell neighbours, as a function of time before swap, in wild-type and *sdk* mutant embryos. Data for graphs can be found at https://doi.org/10.17863/CAM.44798. AP, anteroposterior; GBE, germband extension; Sdk, Sidekick (TIF)

**S9 Fig. (A) Visualisation of how the posterior strain is calculated in our vertex models given the applied stress, $\sigma^{posterior}$.** A small deformation, $E_x$, mapping the $x$-coordinates of vertices as $x \rightarrow x + E_x x$, is applied to the posterior nodes of the tissue in its current configuration. The AP component of the tissue stiffness tensor, $C_{xx}$, can be calculated as the AP component of the change in tissue-level stress over $E_x$. The tissue is then reverted back to its original configuration, and the true posterior strain is calculated as $E^{posterior} = \sigma^{posterior}/C_{xx}$, which is applied by mapping $x$-coordinates of vertices as $x \rightarrow x + E^{posterior}x$. (B) Example of a captured T1 cell rearrangement event in which the junction between cells C and D shortens and is then

replaced by a new junction elongating between cells A and B. We define the resolution phase (grey shading) by the time interval when the shortening and subsequently elongating junctions have a length below 0.75 microns. (C) Cumulative histogram of time spent in the resolution phase for the wild type (blue; n = 1,445) and *sdk* mutant (red; *n* = 990) tissues during 0–30 min of GBE. Exchanges that do not resolve by 30 min of GBE (for example, stuck rosettes and late rearrangements) are excluded. Wild-type cells have a median resolution time of 4 min, whereas *sdk* mutants have a median of 5 min. Kolmogorov–Smirnov test finds significant difference between the distributions ($p < 1.45 \times 10^{-13}$). Data for graph in C can be found at https://doi.org/10.17863/CAM.44798. AP, anteroposterior; GBE, germband extension; Sdk, Sidekick.
(TIF)

## Acknowledgments

We thank Jessica Treisman, Daniel St Johnston, the Bloomington Drosophila Stock center, and the Developmental Studies Hybridoma Bank for fly stocks and antibodies. We thank Jessica Treisman for the information prior to publication that *sdk*[MB5054] was a null allele. We thank Dan Bergstralh (University of Rochester) for hosting Tara Finegan in his laboratory during the completion of this manuscript, Nicola Lawrence (Gurdon Institute, UK) for help with super-resolution imaging, Jenny Evans for technical help, and Claire Lye for feedback on the manuscript.

## Author Contributions

**Conceptualization:** Tara M. Finegan, Nathan Hervieux, Alexander Nestor-Bergmann, Alexander G. Fletcher, Guy B. Blanchard, Bénédicte Sanson.

**Data curation:** Tara M. Finegan, Nathan Hervieux, Alexander Nestor-Bergmann, Guy B. Blanchard.

**Formal analysis:** Tara M. Finegan, Nathan Hervieux, Alexander Nestor-Bergmann, Guy B. Blanchard.

**Funding acquisition:** Bénédicte Sanson.

**Investigation:** Tara M. Finegan, Nathan Hervieux.

**Methodology:** Tara M. Finegan, Nathan Hervieux, Alexander Nestor-Bergmann, Alexander G. Fletcher, Guy B. Blanchard.

**Project administration:** Bénédicte Sanson.

**Resources:** Alexander G. Fletcher, Guy B. Blanchard.

**Software:** Alexander Nestor-Bergmann, Alexander G. Fletcher, Guy B. Blanchard.

**Supervision:** Alexander G. Fletcher, Bénédicte Sanson.

**Validation:** Nathan Hervieux.

**Visualization:** Tara M. Finegan, Nathan Hervieux, Alexander Nestor-Bergmann.

**Writing – original draft:** Tara M. Finegan, Nathan Hervieux, Alexander Nestor-Bergmann, Alexander G. Fletcher, Guy B. Blanchard, Bénédicte Sanson.

**Writing – review & editing:** Tara M. Finegan, Nathan Hervieux, Alexander Nestor-Bergmann, Alexander G. Fletcher, Guy B. Blanchard, Bénédicte Sanson.

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
