## [Editor Report · Decision Letter 0]

22 Jul 2019

Dear Dr Sanson, 

Thank you for submitting your manuscript entitled "The tricellular vertex-specific adhesion molecule Sidekick facilitates polarised cell intercalation during Drosophila axis extension" for consideration as a Research Article by PLOS Biology.

Your manuscript has now been evaluated by the PLOS Biology editorial staff as well as by an Academic Editor with relevant expertise and I am writing to let you know that we would like to send your submission out for external peer review.

**Important**: Please also see below for further information regarding completing the MDAR reporting checklist. The checklist can be accessed here: https://plos.io/MDARChecklist

Please re-submit your manuscript and the checklist, within two working days, i.e. by Jul 24 2019 11:59PM.

Kind regards,

Hashi Wijayatilake, PhD,

Managing Editor

PLOS Biology

INFORMATION REGARDING THE REPORTING CHECKLIST:

PLOS Biology is pleased to support the "minimum reporting standards in the life sciences" initiative (https://osf.io/preprints/metaarxiv/9sm4x/). This effort brings together a number of leading journals and reproducibility experts to develop minimum expectations for reporting information about Materials (including data and code), Design, Analysis and Reporting (MDAR) in published papers. We believe broad alignment on these standards will be to the benefit of authors, reviewers, journals and the wider research community and will help drive better practise in publishing reproducible research. 

We are therefore participating in a community pilot involving a small number of life science journals to test the MDAR checklist. The checklist is intended to help authors, reviewers and editors adopt and implement the minimum reporting framework. 

IMPORTANT: We have chosen your manuscript to participate in this trial. The relevant documents can be located here:

MDAR reporting checklist (to be filled in by you): https://plos.io/MDARChecklist

**We strongly encourage you to complete the MDAR reporting checklist and return it to us with your full submission, as described above. We would also be very grateful if you could complete this author survey:

https://forms.gle/seEgCrDtM6GLKFGQA

Additional background information:

Interpreting the MDAR Framework: https://plos.io/MDARFramework

Please note that your completed checklist and survey will be shared with the minimum reporting standards working group. However, the working group will not be provided with access to the manuscript or any other confidential information including author identities, manuscript titles or abstracts. Feedback from this process will be used to consider next steps, which might include revisions to the content of the checklist. Data and materials from this initial trial will be publicly shared in September 2019. Data will only be provided in aggregate form and will not be parsed by individual article or by journal, so as to respect the confidentiality of responses. 

Please treat the checklist and elaboration as confidential as public release is planned for September 2019.

We would be grateful for any feedback you may have.

---

## [Decision Letter · Decision Letter 1]

20 Aug 2019

Dear Dr Sanson,

Thank you very much for submitting your manuscript "The tricellular vertex-specific adhesion molecule Sidekick facilitates polarised cell intercalation during Drosophila axis extension" for consideration as a Research Article at PLOS Biology. Your manuscript has been evaluated by the PLOS Biology editors, an Academic Editor with relevant expertise, and by several independent reviewers.

The reviewers are generally positive and, in light of their reviews (appended below), we are pleased to offer you the opportunity to address their comments in a revised version that we anticipate should not take you very long. We have discussed the reviews in detail with the Academic Editor and feel that the suggested experiments are not particularly challenging and can be performed in a reasonable time-frame. We will not however insist on Reviewer 2's points 2 and 3 and Reviewer 3's suggested experiment.

Your revisions should address the specific points made by each reviewer. Please submit a file detailing your responses to the editorial requests and a point-by-point response to all of the reviewers' comments that indicates the changes you have made to the manuscript. In addition to a clean copy of the manuscript, please upload a 'track-changes' version of your manuscript that specifies the edits made. This should be uploaded as a "Related" file type. You should also cite any additional relevant literature that has been published since the original submission and mention any additional citations in your response. 

Before you revise your manuscript, please review the following PLOS policy and formatting requirements checklist PDF: http://journals.plos.org/plosbiology/s/file?id=9411/plos-biology-formatting-checklist.pdf. It is helpful if you format your revision according to our requirements - should your paper subsequently be accepted, this will save time at the acceptance stage.

Please note that as a condition of publication PLOS' data policy (http://journals.plos.org/plosbiology/s/data-availability) requires that you make available all data used to draw the conclusions arrived at in your manuscript. If you have not already done so, you must include any data used in your manuscript either in appropriate repositories, within the body of the manuscript, or as supporting information (N.B. this includes any numerical values that were used to generate graphs, histograms etc.). For an example see here: http://www.plosbiology.org/article/info%3Adoi%2F10.1371%2Fjournal.pbio.1001908#s5.

For manuscripts submitted on or after 1st July 2019, we require the original, uncropped and minimally adjusted images supporting all blot and gel results reported in an article's figures or Supporting Information files. We will require these files before a manuscript can be accepted so please prepare them now, if you have not already uploaded them. Please carefully read our guidelines for how to prepare and upload this data: https://journals.plos.org/plosbiology/s/figures#loc-blot-and-gel-reporting-requirements

Upon resubmission, the editors will assess your revision and, assuming the editors and Academic Editor feel that the revised manuscript remains appropriate for the journal, we may send the manuscript for re-review. We aim to consult the same Academic Editor and reviewers for revised manuscripts but may consult others if needed.

We expect to receive your revised manuscript within one month. Please email us (plosbiology@plos.org) to discuss this if you have any questions or concerns, or would like to request an extension. At this stage, your manuscript remains formally under active consideration at our journal; please notify us by email if you do not wish to submit a revision and instead wish to pursue publication elsewhere, so that we may end consideration of the manuscript at PLOS Biology.

When you are ready to submit a revised version of your manuscript, please go to https://www.editorialmanager.com/pbiology/ and log in as an Author. Click the link labelled 'Submissions Needing Revision' where you will find your submission record. 

Sincerely,

Hashi Wijayatilake, PhD, 

Managing Editor

PLOS Biology

REVIEWS:

Reviewer #1: 

Specialized junctional structures at epithelial cell vertices were first described in the 1970s, and more recently a number of their protein components were identified in vertebrates and in Drosophila. However, while several recent studies have focused on tricellular occluding (tight) junctions, which seal the paracellular space at cell vertices in epithelia, essentially no information has been available thus far on the structure and molecular composition of cell vertices at the level of adherens junctions, located adjacent to the occluding junctions. This is an important issue because tricellular junctions (TCJs) are increasingly recognized to play fundamental roles in barrier function, epithelial morphogenesis and tissue homeostasis in development and disease. TCJs integrate mechanical forces in epithelia and act as cell shape sensors that can direct mitotic spindle orientation in dividing epithelial cells. A key prerequisite to understand the dynamics and functions of TCJs in these processes will be to dissect their components, both at the level of the force-transmitting adherens junctions and the occluding tight junctions at cell vertices. 

In this manuscript Finegan et al. describe the subcellular distribution and function of Sidekick (Sdk), a large adhesion molecule with Immunoglobulin and Fibronectin domain repeats that specifically localizes at cell vertices at the level of adherens junctions in various epithelial tissues of Drosophila. The authors use super-resolution imaging to show that Sdk forms string-like structures that appear to extend from the vertex into the adjacent bicellular spaces. Consistent with its predicted role as a homophilic adhesion molecule, Sdk protein is required in pairs of cells for accumulation at tricellular vertices, as the authors demonstrate using genetic mosaic analysis. The distribution of Sdk strings at vertices changes when junctions shrink during cell intercalation events, suggesting that Sdk is involved in this process. 

Surprisingly, sdk function is dispensable for viability. However, the authors demonstrate that sdk is required for cell intercalation during germband elongation in the Drosophila embryo. Careful quantitative analyses of live imaging data revealed that adhesion at vertices between intercalating cells is compromised in sdk mutants, leading to abnormal cell shapes and persisting apical indentations between intercalating cells. Interestingly, loss of sdk affects cell behavior in an anisotropic fashion, with differential effects on anterior-posteriorly (AP) and dorsoventrally (DV) oriented junctions, respectively, resulting in DV shortening and AP elongation of cells during germband elongation. The authors also analyzed the behavior of Sdk in rearranging multicellular rosette structures, which have previously received considerable attention in the field. They show that rosette centers consist of multiple Sdk puncta, arguing that rosettes in fact contain separable apical vertices and that intercalation events within these structures reflect multiple T1 transitions. Together, these new findings represent a significant advance for understanding the dynamic behavior of cells during tissue morphogenesis. 

Finally, the authors employed mathematical modeling to support their experimental findings. The computer simulations (the mathematical basis and validity of which this reviewer is unable to assess) reproduced the cell behaviors observed in sdk mutants, supporting the conclusion that a delay in T1 transitions and altered mechanical properties of the rearranging tissue can explain the observed abnormal cell behavior in sdk mutants. 

Overall, this an excellent piece of work with convincing microscopy data that are carefully quantified and appropriately interpreted. The characterization of the structure and function of apical cell vertices provides an important conceptual advance for the field, with significant implications for epithelial biology in many different contexts and model systems. The work should therefore be of interest to a broad audience of readers. 

Two very recent papers in Developmental Cell describe roles of Sidekick in epithelial morphogenesis and have reached conclusions that are largely similar and complementary to those of Finegan et al. (Letizia et al. (2019). Sidekick Is a Key Component of Tricellular Adherens Junctions that Acts to Resolve Cell Rearrangements. Dev Cell 1–33; 

Uechi and Kuranaga (2019). The Tricellular Junction Protein Sidekick Regulates Vertex Dynamics to Promote Bicellular Junction Extension. Dev Cell 1–27). 

Of note, the work by Finegan et al. is complementary to these two papers, as it provides a more in-depth quantitative analysis of cell intercalation in sdk mutants, and it combines the experimental data with mathematical modeling. 

The authors should address the following (minor) points before the work should be accepted for publication: 

The manuscript needs to be carefully checked for typographical errors. 

Fig. 1A: “G6” should presumably read “M6”

Line 197: There is no cartoon in Fig. 2A (but in Fig. 2B). 

Line 88: change “domain” to “motif”

line 442: change “intracellular PDZ domain” to “intracellular PDZ domain-binding motif”

Fig. 2C-E: To support the authors' interpretations, DV as well as AP oriented junctions should be shown for each stage. 

Line 152: The colocalisation between Sdk-YFP (extracellular) and anti-Sdk (intracellular) staining (Fig. S1) does not strictly rule out the possibility that Sdk protein may be cleaved, because potential cleavage fragments may remain associated. This does not affect the author´s conclusion that Sdk strings are likely contain the entire Sdk protein (Fig. S1F,G), but the wording should be adjusted accordingly. 

Apical “holes” (Fig. 4A): Holes are not evident from the images shown. Can these be shown more clearly using a membrane marker to outline cell surfaces? Also, a cross-sectional view should be shown to better visualize the “holes”. 

The term “hole” suggests that a cell or the tissue is perforated, which does not appear to be the case here. The term apical “indentation” therefore appears more appropriate. 

Line 28, 168: The statement that pair-wise homophilic adhesion is sufficient to enrich Sdk at tricellular vertices is misleading, because the term “sufficient” suggests that bicellular homophilic adhesion alone could provide a mechanism for vertex-specific localization of the protein. The genetic mosaic experiment indicates that two (rather than three) cells contributing Sdk are sufficient for localizing the protein at vertices. However, this does not imply that homophilic adhesion alone mediates vertex localization. Additional mechanisms are likely to be involved. The statement should be rephrased. 

The authors speculate that a sizing mechanism may localize Sdk protein at vertices by excluding it from bicellular spaces and concentrating it at tricellular spaces. They suggest that cell vertices might have larger intercellular spaces than bicellular contacts: This should be readily evident from TEM images. Can the authors show an example to support their hypothesis, or cite a study showing this?

The authors should comment on whether vertebrate homologues of Sdk also localize at cell vertices.

--

Reviewer #2: 

Advance Summary and Potential Significance to Field:

In this manuscript, Finegan, Hervieux et al. study the role of the Ig-containing molecule Sidekick (Sdk) which is specifically localised at tricellular junctions at the level of AJs, in the control of cell shape changes and cell rearrangements during gastrulation movement in early Drosophila embryos. This manuscript extends a previous study made by this team reporting the localisation of Sdk at tricellular junctions, bringing better resolution using super-resolution microscopy approaches, and describes defects in ectodermal cells D/V extension and vertices resolutions during germ band extension (GBE) associated with sdk mutations using elaborate quantitative approaches in live embryos. Finally, adopting their previously published mathematical modelling of GBE cell rearrangements they suggest that the observed defects in sdk mutants are for the most part dependent of delayed vertex resolution.

Comments

The main observations of the paper are the fine description of Sdk localisation at tri-cellular junctions and the identification and fine quantification of a sdk mutant phenotype in the embryonic ectodermal cells where sdk mutations result in defects in cell shape and in vertices and rosette resolution during gastrulation movements. While this observation is interesting, the current study remains however very descriptive and lacks molecular insights to start understanding how Sdk at the tricellular adherens junctions could regulate these morphogenetic events. Below are detailed the main points that would need to be addressed before I can support publication.

1. The phenotype associated with sdk loss-of-function is quantified in great details. However, it is rather weak, since the sdk null mutants are homozygous viable and one could question the specificity of the effects observed which might be unrelated to the sdk gene function, and merely reflect genetic background variations. The authors should thus either provide rescue experiments or at least analyse another independent mutation of sdk, such as sdk∆15, and report similar observations.

As a note here, since sdk null mutants are viable, the defects observed should remain transient and be resolved later in development. Are the differences observed in Figure 3D and 3H on axial shape elongation and DV interface length ever resolved or is the embryo making do with misshapen ectodermal cells?

2. With respect to the sdk mutant phenotype, the observation of the occurrence of holes/gaps in the E-Cadherin and junctions that last longer than in wild-type, is really interesting. Can this phenotype be modified, for instance by lowering the dosage of the shg or arm genes? This would bring more weight to the specificity of the phenotypes reported.

Authors report this defect in particular at the level of rosettes. Even though not explicitly stated, one could thus anticipate that rosettes accumulate over time as gaps at vertices fail to resolve. Authors should thus quantify the number of rosettes over time in wild-type and sdk mutants.

A side note of Figure 4; authors state that holes last longer in sdk mutants, but authors should provide the reader with the information about the relative length of the GBE process in both wild-type and sdk mutants. If the whole development and GBE are slower in sdk mutants, some of the interpretations on persistence of this particular features might have to include that the whole development is slower, and not just the vertices resolution.

3. The current study fails to bring any hints at the molecular mechanisms underlying the sdk mutant phenotype. Given the importance of Myosin planar orientation for some of the features measured here such as DV vs AP axis of junction elongation/shrinkage or rosette formation, and given the importance of Bazooka and of the E-Cadherin complexes for AJ remodelling, authors should study what are the effects of sdk mutations on MyosinII, Bazooka, E-Cad, and alpha- and beta-Catenin distribution either on live or fixed embryos. This is especially important with respect to the E-Cad holes observed in sdk mutants.

4. Finally, while mathematical modelling is provided here, it is not clear how close the different tweaks added and the mathematical model predictions match the observations. Authors should thus show how the different parameters measured in Figure 3 (or at least some of them) and which appear grossly independent from the “delayed vertices resolution” constant that has been modified in the model, are changed and whether they match the experimental data. Authors could also indicate whether the model predicts the perdurance of rosettes as seems to be observed in the experiments (see point 2).

More minor points

A. Authors should try to link more clearly, at least in their interpretations/discussions, the dynamics of Sdk during D/V oriented boundary shrinkage (Figure2) and the phenotypes observed in Figure 3/4/5.

--

Reviewer #3 (Mark Peifer): 

One key question for the cell and developmental biology field is to define mechanisms that allow cells to change shape and move, which requires force generation and major remodeling of cell-cell junctions, without disrupting tissue integrity. Several recent studies have suggested that the actomyosin force-generating structures may be anchored to the cadherin-based cell adhesion machinery at tricellular and multicellular junctions. The authors (and others) have identified a protein, Sidekick, that is highly concentrated at tricellular and multicellular adherens junctions. They combine super-resolution microscopy, genetic analysis, and powerful quantitative and modeling tools to explore the structure of tricellular junctions and the function of Sidekick. I found the results both compelling and exciting. I have some suggestions for a modest set of experiments that would amplify on what is present as well as some suggestions for clarification. This will be of broad interest to cell and developmental biologists.

Suggested experiment

Several proteins have been found that concentrate at tricellular junctions during the stage the authors are studying (though not as completely as Sidekick), and which have known or predicted roles in the process. It would be quite interesting to see super-resolution images of tricellular junctions during germband extension double labeled with Sidekick and non-muscle myosin, Canoe and Enabled.

Data questions and clarifications

The combination of experiments, quantitative analysis and modeling is extremely powerful. However, several of the approaches used are described quite briefly and thus were hard for a simple-minded cell biologist like me to follow. Slight amplifications would allow us to appreciate what was being measured.

1. Fig 5 and associated text. Add a more complete description of the different types of strain rates to go with the nice diagrams.

2. P. 10—this goes through an extremely complex series of measures of intercalation—mostly supplemental. Perhaps focusing in the main text on those which were the most informative?

3. p. 11 what is a Voronoi tiling

The simulations in Figure 6 are a central part of the paper but I felt the authors did not explain them very clearly. My interpretation of these is that the 1st simulation does not match observed tissue strain rates—they should explicitly tell us this and then use the 2nd simulation to add what they now view as an important parameter and walk us through how this improves the match. As part of this they should annotate their resulting cell shape outputs: for example, it’s not obvious that 6G is that different from 6E—point out places where it is and then use additional arrows to point out how 6I is much more similar to what is observed in the mutant

There was one place where I thought there was an overstatement of facts I think needs to be corrected. In both the results and the Discussion the authors make a claim I think needs to be tempered—an example is Discussion-p. 13 “Sidekick, in contrast, is present only at tricellular contacts”. When I examine their superresolution imaging (e.g., Fig. 1D,E) it seems like this is simply not true. It’s highly concentrated there, but there is signal along bicellular borders. This should be toned down.

Introduction

There are a number of places in the Drosophila and mammalian literature where people have explored tension at tricellular and multicellular junctions and proteins that are enriched there. A slightly expanded section on this in the Introduction would help put this work in context. 

Tell us directly in the Introduction that Sidekick loss is non-essential. 

Minor points

p. 5, paragraph 1. Reference figure panels—is it all in S2? Several things in Fig, S1 and S2 could use explanatory arrows (e.g lines 150-152, 167-169

p. 5, paragraph 2. I found it hard to visualize the directionality of the “strings”—X/Y versus Z. Walk us through this a bit more clearly. 

p. 7, lines 186 ff. Rephrase “In this data, Sdk signal along DV-oriented junctions….”

The data in Fig S3A-D is quite interesting and might be moved to a main figure.

PS I loved this idea and saw an intriguing talk at a Gordon Conference this summer that tested the basic mechanism in a simplified model—I wonder if there is an existing literature that might be cited for the underlying principle? “Because of their geometry, vertices in epithelia might have larger intercellular spaces than bicellular contacts. A possibility therefore is that Sdk resides at tricellular contacts because of a sizing mechanism that excludes Sdk from bicellular spaces and concentrate it at tricellular spaces”

---

## [Editor Report · Decision Letter 2]

25 Sep 2019

Dear Dr Sanson,

Thank you for submitting your revised Research Article entitled "The tricellular vertex-specific adhesion molecule Sidekick facilitates polarised cell intercalation during Drosophila axis extension" for publication in PLOS Biology. The Academic Editor has now evaluated the revision and your response to the reviewers. We're pleased to let you know that we're now editorially satisfied with your manuscript. Please do however see below for Data Policy-related requests.

Additionally, before we can formally accept your paper and consider it "in press", we also need to ensure that your article conforms to our guidelines. A member of our team will be in touch shortly with a set of requests. As we can't proceed until these requirements are met, your swift response will help prevent delays to publication.

******* 

Please note that you may have the opportunity to make the peer review history publicly available. The record will include editor decision letters (with reviews) and your responses to reviewer comments. If eligible, we will contact you to opt in or out.

Early Version

To submit your revision, please go to https://www.editorialmanager.com/pbiology/ and log in as an Author. Click the link labelled 'Submissions Needing Revision' to find your submission record. Your revised submission must include a cover letter and a track-changes file indicating any changes that you have made to the manuscript. 

Sincerely,

Hashi Wijayatilake, PhD, 

Managing Editor

PLOS Biology

DATA POLICY:

You are aware of the PLOS Data Policy, which requires that all data be made available without restriction: http://journals.plos.org/plosbiology/s/data-availability. For more information, please also see this editorial: http://dx.doi.org/10.1371/journal.pbio.1001797

Figs. 2FGH, 4C-H, 5DEF, 6B-G, 7D-J, S4A-D, S5, S7, S8, S9

ALSO:

- Please update your Data Availability Statement (in the submission form) to remove the 'if the paper is accepted...' sentence and just refer to the source data file/s.

- Please provide legends for the supplementary methods S1 Text file and the S1 Table.

- Please also ensure that figure legends in your manuscript include information on where the underlying data can be found.

For manuscripts submitted on or after 1st July 2019, we require the original, uncropped and minimally adjusted images supporting all blot and gel results reported in an article's figures or Supporting Information files. We will require these files before a manuscript can be accepted so please prepare them now, if you have not already uploaded them. Please carefully read our guidelines for how to prepare and upload this data: https://journals.plos.org/plosbiology/s/figures#loc-blot-and-gel-reporting-requirements.

---

## [Editor Report · Decision Letter 3]

31 Oct 2019

Dear Dr Sanson,

On behalf of my colleagues and the Academic Editor, Nicolas Tapon, I am pleased to inform you that we will be delighted to publish your Research Article in PLOS Biology. 

Early Version

PRESS 

Kind regards,

Hannah Harwood

Publication Assistant, 

PLOS Biology

on behalf of

Hashi Wijayatilake,

Managing Editor

PLOS Biology